# Stratification of enterochromaffin cells by single-cell expression analysis

Yan Song[1,2,3†], Linda J Fothergill[4,5†], Kari S Lee[1,2,3], Brandon Y Liu[1,2,3], Ada Koo[4], Mark Perelis[1,2,3], Shanti Diwakarla[4], Brid Callaghan[4], Jie Huang[6], Jill Wykosky[6], John B Furness[4,5]*, Gene W Yeo[1,2,3]*

[1]Department of Cellular and Molecular Medicine, University of California San Diego, La Jolla, United States; [2]Stem Cell Program, University of California San Diego, La Jolla, United States; [3]Institute for Genomic Medicine, University of California San Diego, La Jolla, United States; [4]Department of Anatomy & Physiology, University of Melbourne, Parkville, Australia; [5]Florey Institute of Neuroscience and Mental Health, Parkville, Australia; [6]Takeda Pharmaceuticals, San Diego, United States

## eLife Assessment

This **important** study presents a transcriptomic analysis of enterochromaffin cells in the intestine. The evidence supporting the authors' claims is **solid**, although the functional analysis is focused on the Piezo2-expressing subset in the colon. The work will be of interest to biologists working on intestinal mucosal biology.

**\*For correspondence:**
j.furness@unimelb.edu.au (JBF);
geneyeo@ucsd.edu (GWY)

†These authors contributed equally to this work

**Abstract** Dynamic interactions between gut mucosal cells and the external environment are essential to maintain gut homeostasis. Enterochromaffin (EC) cells transduce both chemical and mechanical signals and produce 5-hydroxytryptamine to mediate disparate physiological responses. However, the molecular and cellular basis for functional diversity of ECs remains to be adequately defined. Here, we integrated single-cell transcriptomics with spatial image analysis to identify 14 EC clusters that are topographically organized along the gut. Subtypes predicted to be sensitive to the chemical environment and mechanical forces were identified that express distinct transcription factors and hormones. A *Piezo2*+ population in the distal colon was endowed with a distinctive neuronal signature. Using a combination of genetic, chemogenetic, and pharmacological approaches, we demonstrated *Piezo2*+ ECs are required for normal colon motility. Our study constructs a molecular map for ECs and offers a framework for deconvoluting EC cells with pleiotropic functions.

## Introduction

The capacity of the gut epithelium to sense and react to its surrounding environment is essential for proper homeostasis. Enteroendocrine (EEC) cells within the gut epithelium respond to a wide range of stimuli, such as dietary nutrients, irritants, microbiota products, and inflammatory agents by releasing a variety of hormones and neurotransmitters to relay sensory information to the nervous system, musculature, immune cells, and other tissues (*Gribble and Reimann, 2016*; *Gribble and Reimann, 2019*). In particular, enterochromaffin (EC) cells represent one of the major epithelial sensors. Historically, EC cells were histologically identified as the first type of gastrointestinal endocrine cells and have been thought of as a single-cell type for about seven decades, until the emergence of recent studies that point to their heterogeneity (*Erspamer and Asero, 1952*; *Berger et al., 2009*; *Gershon, 2013*; *Diwakarla et al., 2017*; *Martin et al., 2017b*).

EC cells constitute less than 1% of the total intestinal epithelium cells, but they produce >90% of the body's 5-hydroxytryptamine (5-HT, serotonin) to modulate a wide range of physiological functions (*Sjölund et al., 1983*; *Berger et al., 2009*; *Gershon, 2013*; *Diwakarla et al., 2017*; *Martin et al., 2017b*). Dysregulation of peripheral 5-HT levels is implicated in the pathogenesis of gastrointestinal (GI) diseases (*Coleman et al., 2006*; *Di Sabatino et al., 2014*), cardiovascular disease (*Ramage and Villalón, 2008*), osteoporosis (*Ducy and Karsenty, 2010*), and are associated with sudden infant death syndrome (*Haynes et al., 2017*) as well as psychiatric disorders, including autism spectrum disorders (*Anderson et al., 1990*; *Muller et al., 2016*). The distinct and highly diverse functions of peripheral 5-HT suggest the possibility of specialization of EC subtypes that react to specific stimuli, such as chemicals in the lumen of the gut, mechanical forces, dietary toxins, microbiome metabolites, inflammatory mediators, and other GI hormones.

Since EC cells are infrequent and distributed throughout the gut wall, traditional approaches have utilized endocrine tumor cell lines, whole tissue preparations, or genetic models (such as tryptophan hydroxylase 1 knock-out) to investigate the functions of EC cells, which have generally assumed that EC cells are a single-cell type and have not addressed their heterogeneity in sensory modalities. A recent study exploited intestinal organoids and described EC cells as polymodal chemical sensors, but lacked the resolution to disentangle the origin of the polymodality (*Bellono et al., 2017*). Some studies have compared small intestinal and colonic EC cells by RT-PCR (*Martin et al., 2017a*; *Lund et al., 2018*), and one study used single-cell RNA sequencing (scRNA-seq) to compare a small number of human EC cells from the stomach and duodenum (*Busslinger et al., 2021*). Single-cell transcriptomics have been utilized to profile intestinal epithelia (*Haber et al., 2017*; *Wang et al., 2020*) and EEC cells (*Glass et al., 2017*; *Gehart et al., 2019*; *Billing et al., 2019*), which include EC cells, however, questions regarding regional, molecular, and functional heterogeneity of EC cells remain to be investigated in depth.

Here, we generated a genetic reporter of tryptophan hydroxylase 1 (Tph1), the rate-limiting enzyme of 5-HT synthesis in EC cells, and applied scRNA-seq to profile >6000 EC cells. Together with spatial imaging analysis at single-cell resolution, we identified 14 clusters of EC cells distributed along the rostro-caudal and crypt–villus axes of the gut. We stratified EC subsets based on their repertoires of sensory molecules. In particular, we demonstrate an important role of the $Piezo2^+/Ascl1^+/Tph1^+$ subpopulation in normal gut motility, one of the proposed functions of EC-derived 5-HT (*Bulbring and Crema, 1958*; *Bulbring and Crema, 1959*; *Martin et al., 2022*; *Wei et al., 2021*). Our comprehensive molecular resource and findings provide direct evidence of molecular and cellular heterogeneity of EC cells and is anticipated to be valuable for future studies.

## Results

### Generation and characterization of a *Tph1-bacTRAP* reporter model

To systematically analyze EC cells we generated a *Tph1*-bacTRAP mouse strain by placing a *Rpl10-GFP* fusion gene under the transcriptional control of the *Tph1* gene in a BAC construct (*Figure 1*, *Figure 1—figure supplement 1a*, *Figure 1—source data 1*). In the *Tph1*-bacTRAP line that we generated, all GFP$^+$ (representing Tph1$^+$) cells in the duodenum and over 95% of the cells in the jejunum and large intestine were immunoreactive for 5-HT (*Figure 1a, b*). Cells that stained positive for 5-HT but negative for GFP were also observed (*Figure 1b*), which are likely to include tuft cells that store but do not synthesize 5-HT (*Figure 1—figure supplement 1b*; *Cheng et al., 2019*). Epithelial cells from the duodenum, jejunum, ileum, and colon were isolated from the *Tph1*-bacTRAP mice (*Figure 1c*, *Figure 1—figure supplement 1c, d*), sorted via fluorescence-activated cell sorting (FACS) and both GFP$^+$ and GFP$^-$ cells were subjected to scRNA-seq. Among a total of 4729 cells, *Tph1* transcripts were measured in 88.9% of GFP$^+$ cells, together with the chromogranin genes, *Chga* (in 97.3% of GFP$^+$ cells) and *Chgb* (in 98.8% of GFP$^+$ cells), established markers for EC cells (*Figure 1—figure supplement 1e*). Cluster analysis indicated that 23% of GFP$^+$ cells were grouped with GFP$^-$ cells (*Figure 1—figure supplement 1d*), although these cells had higher levels of the EC marker genes compared to the GFP$^-$ cells they clustered with (*Figure 1—figure supplement 1e, f*). It is possible that some GFP is expressed in cells that have not yet fully committed to the EC lineage, or that there is some expression in cells outside this lineage, for example, in mast cells. Given the small sample size, we did not further

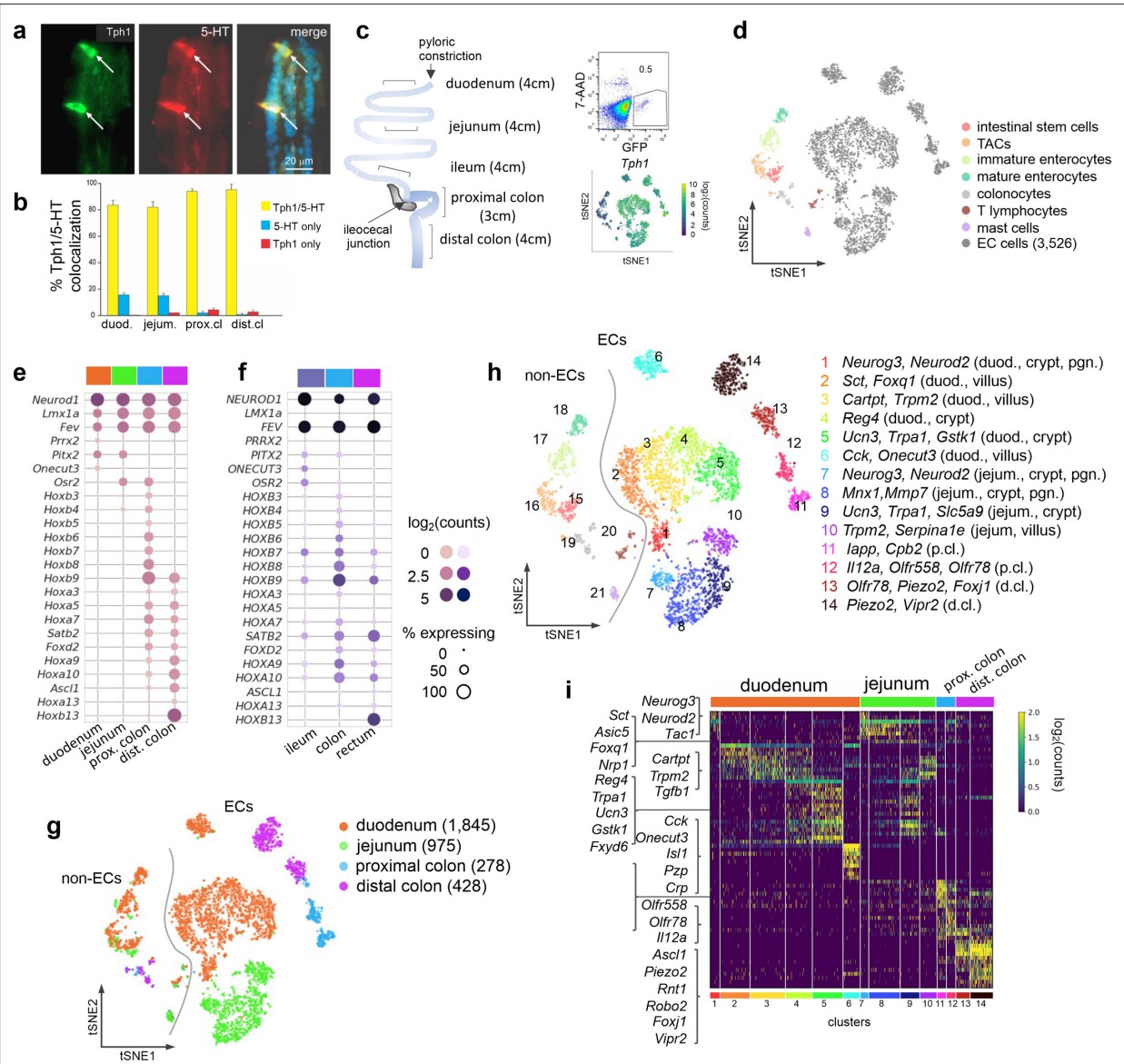

**Figure 1.** scRNA-seq identifies distinct intestinal EC cell clusters. Dual IF staining of 5-hydroxytryptamine (5-HT) and GFP (representing Tph1) (**a**) and their quantification/colocalization in the indicated regions of *Tph1*-bacTRAP mice (**b**). Schematic showing the isolation and enrichment of GFP⁺ cells from the indicated regions of the GI tract for scRNA-seq (**c**). *Upper right*: Fluorescence-activated cell sorting (FACS) plot of dissociated gut epithelial cells from *Tph1*-bacTRAP mice. Gate shows GFP⁺ cells, which account for ~0.5% of total viable gut epithelial cells (7-AAD⁻ cells). *Lower right*: t-SNE projection of all GFP⁺ cells superimposed on an expression heatmap of *Tph1* in GFP⁺ cells. t-SNE projection of all GFP⁺ cells isolated from *Tph1*-bacTRAP mice in the second cell profiling experiment, from which 3526 EC cells were identified and subjected to further analysis (**d**). TACs: transit amplifying cells; EC cells: enterochromaffin cells. (**e, f**) Transcription factors (TFs) that are differentially expressed in the mouse EC cells isolated from *Tph1*-bacTRAP mice along the rostro-caudal axis presented by regions as indicated in the color bar (**e**). Expression data of the human orthologues of the same TFs were extracted from human gut mucosa dataset (GSE125970). Enteroendocrine (EEC) cells were selected and presented by regions (**f**). Size of the circles represents percentage of expression and the intensity of the circles represents aggregated expression of indicated TFs in cells partitioned by regions. (**g**) t-SNE projection of all GFP⁺ cells color-coded by their regions in the GI tract. Numbers of cells retained from each region are indicated in parentheses. Dashed line demarcates the separation of non-EC cells (including stem cells, TACs, immature enterocytes, mature enterocytes, colonocytes, T lymphocytes, and mast cells) versus EC cells. (**h**) tSNE projection of all GFP⁺ cells color-coded by clusters that were identified via the Louvain method. Fourteen clusters of EC cells (clusters 1–14) and 7 clusters of non-EC cells were identified (clusters 15–21, as described in (**d**)). Key marker genes are listed for each cluster. duod.: duodenum, jejum.: jejunum, pgn: progenitor. (**i**) Heatmap of the top 5–10 signature genes for each cluster presented as normalized log₂(counts) in all EC cells (in columns). Color-coded bar at the bottom represents the clusters identified in (**h**).

The online version of this article includes the following source data and figure supplement(s) for figure 1:

**Source data 1.** Expression data for specific detected genes.

**Figure supplement 1.** scRNA-seq identifies distinct intestinal enterochromaffin (EC) cell clusters.

investigate these cells in this dataset. In *Figure 1—figure supplement 1d, f*, we refer to the GFP+ cells that clustered with the GFP− cells as 'non-EC cells'.

An independent single-cell profiling experiment was performed focusing on GFP+ cells (0.3–0.5% of total dissociated epithelial cells) from the duodenum, jejunum, and proximal and distal colon of the *Tph1*-bacTRAP mice, where numbers of GFP+ cells were adequate (i.e., excluding the ileum) (*Figure 1d*). 4348 high-quality single cells were obtained, of which 19% comprised of non-EC cells and identified as stem cells, transit amplifying cells (TACs), immature enterocytes, mature enterocytes, colonocytes, T lymphocytes, and mucosal mast cells based on their respective markers (*Haber et al., 2017*; *Figure 1d*, *Figure 1—figure supplement 1g*). It is possible that the stem cell and TAC clusters include cells that are in the process of differentiating into EC cells. However, given that they have not fully committed to the lineage, we do not consider it appropriate to classify them as 'EC cells' for the purposes of analyzing EC cell types in this study. A total of 3526 EC cells (at threshold >500 detected genes and <10% mitochondrial transcripts) were retained for analysis.

## Distinct EC subpopulations along the rostro-caudal axis

EC cells are one of the few EEC cell types distributed along the full length of the GI tract. The most significant transcriptomic distinction was observed between small intestinal and colonic EC cells as revealed by principal component analysis (PCA) (*Figure 1—figure supplement 1h*), even though all the EC cells expressed a core set of EC markers (*Figure 1—figure supplement 1i*). Unsupervised hierarchical clustering complemented with bootstrap resampling partitioned EC single cells by regions based on their overall transcriptomic similarity (*Figure 1—figure supplement 1j*). The regional distinction of EC cells is apparent from the examination of transcription factors (TFs) along the rostro-caudal axis. While *Pitx2* and *Osr2* demonstrated preferential enrichment in different segments of the gut, a suite of *Hox* genes were only observed in the colon, with *Hoxb13* specifically detected in the distal colon (*Figure 1e*). This pattern was shared by all gut epithelial cells (*Figure 1—figure supplement 1k*) and was largely conserved in the human gut mucosa based on our comparative analysis of a scRNA-seq dataset of biopsy samples from human ileum, colon and rectum (*Wang et al., 2020*). Notably, OSR2 and HOXB13 were preferentially enriched in the ileum and rectum, respectively, in these human samples (*Figure 1f*). Consistent with a previous study, *Olfr78* and *Olfr558* were enriched in the colon but not the small intestine (*Lund et al., 2018*).

To investigate the diversity of EC cells within each intestinal region, we clustered them using the Louvain method for unsupervised community detection and resolved 14 EC clusters that were mostly demarcated by regions: duodenum (clusters 1–6), jejunum (clusters 7–10), proximal colon (clusters 11–12), and distal colon (clusters 13–14, *Figure 1g, h*). The EC cells from either duodenum (clusters 1–5) or jejunum (clusters 7–10) displayed a continuum in t-distributed stochastic neighbor embedding (tSNE) space, indicating a gradual transcriptomic change in the SI, whereas colonic EC cells formed clusters distinct from the SI clusters (*Figure 1g, h*). Interestingly, cluster 6 (duodenum EC cells) was projected away from the rest of the SI EC cells in the tSNE space, suggesting a distinct molecular profile of the cells (see below).

## SI EC cells are predicted to switch sensors and hormone compositions along the crypt–villus axis

Next, we resolved the identities of the EC clusters using both known and newly identified marker genes (*Figure 1h, i*). Since all intestinal EEC cells, including EC cells, derive from Neurog3+ cells, we annotated the *Neurog3*+ clusters as EC progenitors in the duodenum (cluster 1) and jejunum (cluster 7). Lineage tracing studies have established that crypt and villus EC cells preferentially express *Tac1* (encoding tachykinin precursor 1) and *Sct* (encoding secretin), respectively (*Gehart et al., 2019*; *Roth and Gordon, 1990*; *Beumer et al., 2018*). We annotated clusters 4/5 (duodenum) and 8/9 (jejunum) as crypt clusters and clusters 2/3/6 (duodenum) and 10 (jejunum) as villus clusters (*Figure 2a*) based on the relative expression levels of *Tac1* and *Sct* (validated in *Figure 2—figure supplement 1b-e*). We note that this division is not precise, but reflects a gradient between cells in the crypts and villi.

Since a spatial transcriptomic zonation along the crypt–villus axis had been demonstrated in intestinal enterocytes (*Moor et al., 2018*) and EEC cells (*Gehart et al., 2019*), we further investigated whether EC cells diversify their transcriptome along the same axis to be poised for various functions. Using differential expression (DE) analysis, we identified a set of signature genes that

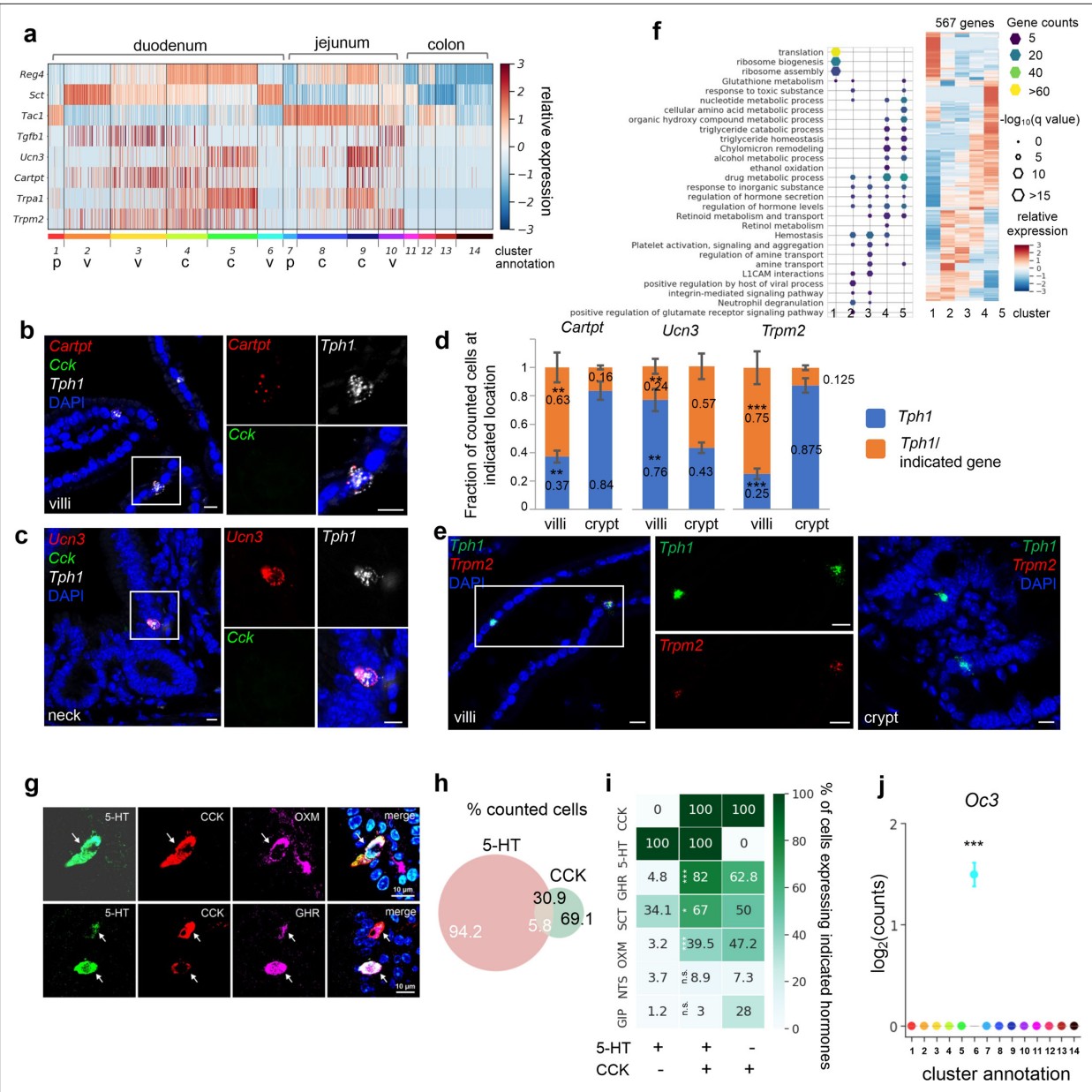

**Figure 2.** SI enterochromaffin (EC) cells are predicted to switch sensors and hormone compositions along the crypt–villus axis. (**a**) Heatmap of representative genes with differential expression patterns between clusters annotated as crypt or villus. Relative gene expression (*z*-score) is shown across all single EC cells. Color-coded bar at the bottom represents the clusters. p: progenitor, v: villus, c: crypt. smRNA-FISH of *Cartpt*, *Tph1*, and *Cck* (**b**); *Ucn3*, *Tph1*, and *Cck* (**c**); or *Trmp2* and *Tph1* (**e**). The boxed regions are enlarged on the right and split into individual channels. Data shown are representative examples from three independent mice. Scale bars: 10 μm. (**d**) Quantitation of *Cartpt*, *Ucn3*, and *Trmp2* positivity in the *Tph1*[+] cells in the crypt versus villus. For each quantitation, 130–160 *Tph1*[+] cells were counted from the crypt or villus of the duodenum in three independent mice. **p < 0.01, ***p < 0.001; unpaired two-tailed Student's *t*-test comparing the fractions in the villi against those in the crypt. (**f**) Gene ontology analysis of the 567DE genes identified in the duodenal clusters. DE genes were determined by false discovery rate (FDR) <10[−10] against every other cluster (among clusters 1–5) based on Wilcoxon rank sum test and corrected by the Benjamini–Hochberg procedure. Relative expression (*z*-score) of DE genes is shown on the right, GO analysis of DE genes on the left. Size of the hexagons represents the *q*-value of enrichment after −log[10] transformation, and the density represents the number of genes per GO term. Accumulative hypergeometric testing was conducted for enrichment analysis and Sidak–Bonferroni correction was applied to correct for multiple testing. Cells co-expressing 5-HT, CCK, and a third hormone product shown by IF staining, such as a *Gcg* product (GHR), oxyntomodulin (OXM, upper panel) and the *Ghrl* product, ghrelin (GHR, lower panel). Arrows point to the triple-positive cells. Note that the relative levels of hormones vary considerably among cells, as shown in the 5-HT, CCK, and ghrelin triple stain. Venn diagram showing co-expression of 5-HT and CCK based on IF staining with indicated antibodies. Numbers represent percentages out of all 5-HT[+] cells (white) or CCK[+] cells (black). Summary of 5-HT or CCK single-positive cells and 5-HT/CCK double-positive cells (by columns) producing a third hormone (by rows) as identified by IF staining. Heatmap and annotated numbers represent the percentage out of all cells in individual columns. ***p < 10[−10], **p < 10[−2] by hypergeometric

*Figure 2 continued on next page*

*Figure 2 continued*

tests against the 5-HT single-positive cells. (**g–i**) are based on IF staining for indicated peptides/hormones in three different animals. Point plot depicting the median expression of *Oc3* in each cluster showing enrichment of *Oc3* expression in cluster 6 (*Cck*+/*Tph1*+ cells) from the *Tph1-bacTRAP* dataset. Error bars represent upper and lower quantiles. ***p < 10$^{-50}$; two-tailed Kolmogorov–Smirnov statistic between the observed Oc3 distribution versus the bootstrap-facilitated randomization control distribution (median of *n* = 500 randomizations shown in gray boxplots).

The online version of this article includes the following figure supplement(s) for figure 2:

**Figure supplement 1.** SI enterochromaffin (EC) cells are predicted to switch sensors and hormone compositions along the crypt–villus axis.

were preferentially enriched in EC clusters annotated as being from the villus or crypt (*Figure 2a*, *Figure 2—figure supplement 1f*). Their differential distribution was further tested to be statistically significant by comparing to bootstrap-facilitated randomization of gene subsets (*Figure 2—figure supplement 1g,h*).

To validate the spatial distribution of candidate genes, we utilized single-molecule RNA-FISH (smRNA-FISH) as an orthogonal method. We found that molecular sensors, such as *Trpa1* and *Trpm2*, together with additional hormone peptides, such as *Ucn3* and *Cartpt*, were differentially distributed in the EC cells along the crypt–villus axis. To illustrate, scRNA-seq suggested that *Ucn3* (encoding urocortin3) was frequently observed in the crypt clusters 4/5, whereas *Cartpt* (encoding cocaine and amphetamine regulated transcript prepropeptide) was enriched in the villus cluster 3 and to a lesser degree in cluster 4 (*Figure 2a*). Consistently, smRNA-FISH demonstrated that *Ucn3* and *Cartpt* were selectively co-expressed with *Tph1*, but not with *Cck* (encoding cholecystokinin; expressed in cluster 6 as discussed below) (*Figure 2b, c*). Specifically, *Ucn3* was found in 57.0% (±3.3%) of *Tph1*+ cells at the crypt or the neck of the villus, but observed in 24.0% (±2.1%) of *Tph1*+ at the villus. In contrast, *Cartpt* was found in 63% (±3.5%) and 16% (±0.5%) of *Tph1*+ cells in the villus and crypt, respectively (*Figure 2b–d*). Another pair of examples is the phytochemical sensor *Trpa1* (encoding transient receptor potential cation channel, subfamily A, member 1) and a novel sensor gene *Trpm2* (encoding transient receptor potential cation channel, subfamily M, member 2). *Trpa1* was frequently detected in the crypt cluster 5 in scRNA-seq, in agreement with its crypt location previously reported in rodent and human intestine (*Cho et al., 2014*; *Nozawa et al., 2009*), whereas *Trpm2* cells were mostly distributed in the villus clusters based on scRNA-seq analysis and further validated by smRNA-FISH to be co-expressed with 75% (±4.1%) and 12.5% (±0.4%) of *Tph1*+ cells in the villi and crypt, respectively (*Figure 2d, e*). Taken together, our findings are suggestive of a concomitant signature switch in the EC cells from *Tac1/Unc3/Trpa1* to *Sct/Cartpt/Trpm2* as the cells migrate from the crypt to the villus. In addition, scRNA-seq indicated that genes associated with oxidative detoxification including peroxidases and oxygenases (e.g., *Gstk1*, *Alb*, *Fmo1*, and *Fmo2*) were preferentially enriched in the crypt clusters 5/4, in contrast to the villus clusters 2/3 (*Figure 2—figure supplement 1f*). Furthermore, *Reg4*, *Ucn3*, *Trpa1*, *Gstk1*, and *Fmo1* expression levels are very low in progenitor clusters 1 and 7, which is consistent with reports from a time-resolved lineage tracing model that suggest that these genes are expressed late in the differentiation process of EEC cells (*Gehart et al., 2019*).

To assess the functional states of the duodenal and jejunal EC clusters, we performed gene ontology (GO) enrichment analysis based on cluster-enriched genes (identified as log$_2$(fold-change) >2, false discovery rate [FDR] <10$^{-10}$) (*Figure 2f*). Consistent with its annotation as a progenitor cluster, cluster 1 was enriched with terms 'translation' and 'ribosome biogenesis', suggestive of a high protein production state. Crypt clusters 4/5 were enriched with terms related to metabolism and hydroxy compound/alcohol metabolic process/detoxification, whereas villus clusters 2/3 were enriched with terms 'hemostasis', 'viral process', and 'neutrophil degranulation', suggesting the villus clusters may be involved in host defense (also see below for cluster 6).

## *Cck*, *Oc3*, and *Tph1* identify an EC subpopulation with a dual sensory signature

Emerging data suggest considerable co-expression of hormones within individual EEC cells, including EC cells (*Gribble and Reimann, 2016*; *Diwakarla et al., 2017*; *Billing et al., 2019*; *Gehart et al., 2019*). Among all the EC clusters, the largest number of hormone-coding genes was observed in cluster 6 expressing the highest levels of *Sct*, *Cck*, and *Ghrl*, followed by *Gcg* and *Nts* (*Figure 2—figure supplement 1i*). Cluster 6 was almost exclusively constituted of duodenal EC cells (198/203,

97.5%) and comprised 10.9% of all retained duodenal EC cells (*Figure 3—figure supplement 1*), thus representing <0.1% of total duodenal epithelial cells. To validate this small population, we investigated the co-expression of hormonal products by immunohistochemistry in the duodenum and found that 5.8 (±1.4%) of 5-HT expressing cells were positive for CCK, representing 30.9 (±6.6%) of CCK-positive cells (*Figure 2g–i*). Notably, of the 5-HT/CCK double-positive cells, a large proportion demonstrated positivity for ghrelin (GHR, 82%), oxyntomodulin (OXM, 39.5%), one of the peptide products of the pre-proglucagon gene (*Gcg*), neurotensin (NTS, 8.9%), and a small percentage was positive for glucose-dependent insulinotropic polypeptide (GIP, 3.0%). These percentages were significantly reduced in 5-HT$^+$/CCK$^−$ cells, which presented 4.8%, 3.2%, 3.7%, and 1.2% positivity for GHR, OXM, NTS, and GIP, respectively (*Figure 2i*). Thus, cluster 6 represents a subpopulation of EC cells with a broader spectrum of hormones (referred to as *Cck$^+$/Tph1$^+$* hereafter).

In a survey of TFs (*Gribble and Reimann, 2016*; *Gehart et al., 2019*) that potentially specify the *Cck$^+$/Tph1$^+$* cells, we found *Onecut3* (*Oc3*) to be highly enriched in cluster 6 cells (*Figure 2j*, *Figure 2—figure supplement 1*), which we validated in an independent scRNA-seq dataset of *Neurod1$^+$* EEC cells (*Figure 3i, j*, *Figure 3—figure supplement 1*, see below). Using smRNA-FISH, we found that 100% of the *Cck$^+$/Tph1$^+$* cells were positive for *Oc3*, whereas only 11.4% (±2.1%) of *Cck$^-$/Tph1$^+$* cells and 58% (±3.7%) of *Cck$^+$/Tph1$^−$* cells were positive for *Oc3* (*Figure 3a–c*, *Figure 3—source data 1*). *Oc3* single-positive cells were rarely observed (1.8% ± 2.4% of 970 counted cells examined for *Cck*, *Oc3*, and *Tph1*; *Figure 3b, d*), indicating that *Oc3* is largely restricted to *Cck$^+$* and/or *Tph1$^+$* populations and may be associated with the specification of *Cck$^+$/Tph1$^+$* cells. Notably, triple-positive cells (*Cck$^+$/Oc3$^+$/Tph1$^+$*) were more frequently observed in the villus (25 out of every 100 counted single-, double-, or triple-positive cells) than in the crypt (6 out of every 100 single-, double-, or triple-positive cells; *Figure 3d*). Focusing on *Oc3$^+$* cells identified via smRNA-FISH, we found that the fraction of *Cck$^+$/Oc3$^+$* cells decreased from 69.3% (±0.98%) in the crypt to 22.8% (±5.3%) in the villus, while *Tph1$^+$/Oc3$^+$* cells increased from 17.3% (±2.4%) in the crypt to 38.0% (±3.2%) in the villus (*Figure 3d*), suggesting a likelihood that *Cck$^+$/Oc3$^+$* double-positive cells gradually acquire the ability to generate *Tph1* transcripts during their migration to the villus. Supporting this, triple-positive cells (*Cck$^+$/Oc3$^+$/Tph1$^+$*) were found to express TFs shared with other EEC cells, including *Etv1*, *Isl1*, and *Arx*, but displayed lower levels of *Lmx1α* enriched in the rest of *Tph1$^+$* cells (*Gross et al., 2016*; *Figure 3—figure supplement 1*).

We further determined the molecular signature unique to *Cck$^+$/Oc3$^+$/Tph1$^+$* cells by contrasting them to the rest of the EC cells (*Figure 3e*), or other EEC cells (*Figure 3j*, *Figure 3—figure supplement 1*). Specifically, discrete expression of *Crp* (encoding C-reactive protein) was identified in cluster 6 by scRNA-seq and mapped to *Cck$^+$/Tph1$^+$* cells by smRNA-FISH (*Figure 3g*). Crp is a secreted bacterial pattern-recognition receptor involved in complement-mediated phagocytosis, known to be secreted by hepatocytes in response to inflammatory cytokines during infection or acute tissue injury (*Black et al., 2004*). Similarly, several genes encoding molecules recognizing pathogen-associated molecular patterns were found in cluster 6 cells, including *Tril* (encoding TLR4 interactor with leucine rich repeats), *Tlr2* (encoding Toll-like receptor 2) and *Tlr5* (encoding Toll-like receptor 5). We validated the latter two by smRNA-FISH to be specific in *Cck$^+$/Tph1$^+$* cells (*Figure 3—figure supplement 1*), supporting the notion that these cells play a role in pathogen or toxin recognition. Concordantly, *Bcam*, encoding a cell surface receptor that recognizes a major virulence factor (CNF1) of pathogenic *E. coli* (*Piteau et al., 2014*), was enriched in cluster 6 (*Figure 3e*). Along with cluster 6, the two villus clusters (cluster 2/3) were enriched with GO terms associated with host defense (*Figure 2f*), suggesting a continuous evolution of EC cells along the crypt–villus axis to specify a complex subpopulation that mediates acute responses to tissue challenge. In addition to the unique signature of pathogen/toxin recognition, genes encoding G-protein-coupled receptors (GPCRs) associated with nutrient sensing and energy homeostasis, *Mc4r* and *Casr*, were distinctly identified in cluster 6 cells (*Figure 3e*). Mc4r (melanocortin receptor 4, encoded by *Mc4r*) plays a central role in energy homeostasis and satiety (*Huszar et al., 1997*; *Lotta et al., 2019*). CaR (calcium-sensing receptor, encoded by Casr) acts as a sensor for extracellular calcium and aromatic amino acids (*Conigrave et al., 2000*). We further validated expression of *Mc4r* and *Casr* by smRNA-FISH to be specific to *Cck$^+$/Tph1$^+$* cells but not single-*Tph1$^+$* or -*Cck$^+$* cells (*Figure 3h*, *Figure 3—figure supplement 1*). Although the nutrient sensing GPCRs were observed at relatively lower levels and frequencies in scRNA-seq data (*Figure 3e, j*), among the *Mc4r$^+$* or *Casr$^+$* cells, >85% of them co-expressed at least one of the genes associated

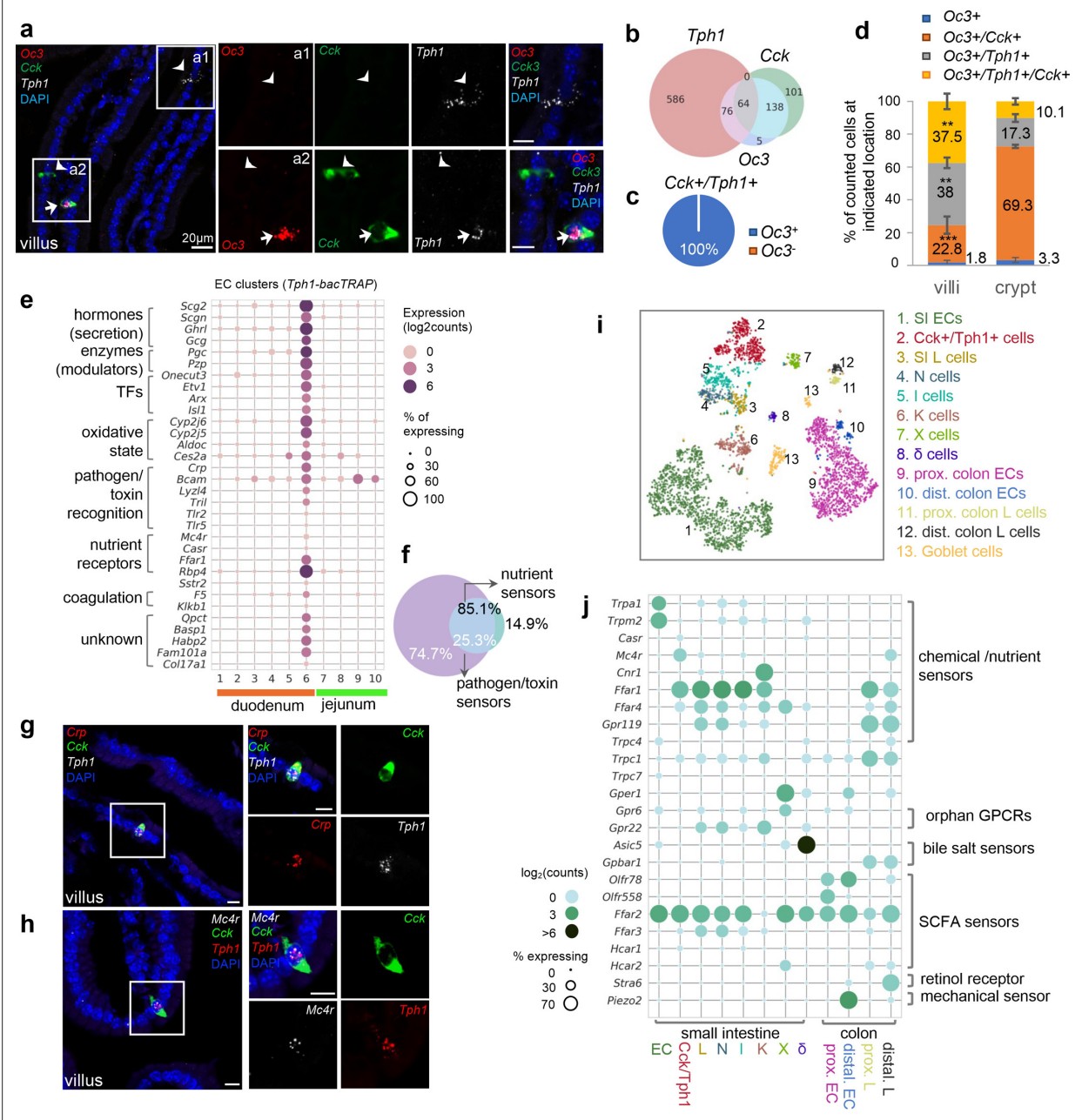

**Figure 3.** *Cck*, *Oc3*, and *Tph1* specify an enterochromaffin (EC) subpopulation with a dual sensory signature. (**a**) smRNA-FISH of *Cck/Oc3/Tph1*. Two boxed regions (**a1, a2**) are enlarged on the right and split into individual channels. An arrow points to a *Cck⁺/Oc3⁺/Tph1⁺* cell in the villi and arrowheads point to *Tph1* or *Cck* single-positive cells with no *Oc3* expression in the villi. Images are representative from four different animals. Scale bars: 20 µm. (**b**) Venn diagram showing overlaps of *Tph1-*, *Cck-*, and *Oc3-*expressing cells based on smRNA-FISH performed on duodenal sections. Numbers of counted cells are annotated based on four sections per mouse in four different animals. (**c**) Pie chart showing all counted *Cck⁺/Tph1⁺* cells partitioned by *Oc3* positivity based on smRNA-FISH. (**d**) Quantitation of *Oc3* expressing cells by smRNA-FISH in the villi versus in the crypts. **p < 0.01, ***p < 0.001, unpaired two-tailed Student's *t*-test comparing the percentages in the villi against those in the crypts. (**e**) Signature genes identified in *Cck⁺/Oc3⁺/Tph1⁺* cells by scRNA-seq in the *Tph1-*bacTRAP dataset. Size of the circles represents percentage of expression and intensity of the circles represents aggregated expression of indicated genes. (**f**) Venn diagram showing co-expression of two sets of genes summaried by the aggregated expression of genes associated with pathogen/toxin recognition (*Crp*, *Lyz4*, *Tril*, *Tlr2*, and *Tlr5*) versus genes associated with nutrient sensing and homeostasis (*Mc4r* and *Casr*) in the *Tph1-*bacTRAP dataset. smRNA-FISH of *Crp*, *Cck*, and *Tph1* (**g**) and *Mc4r*, *Cck*, and *Tph1* (**h**). The boxed regions are enlarged on the right and split into individual channels. Data are representative from three different animals. Scale bars: 10 µm. tSNE representation of single *Neurod1-tdTomato⁺* cells isolated from the gut. Cells are color-coded by clusters that were identified via the Louvain method. Dot plot of genes coding G-protein-coupled receptors (GPCRs), TRP channels, SLC transporters, purinergic receptors, and prostaglandin receptors, identified in clusters as shown

*Figure 3 continued on next page*

*Figure 3 continued*

in (**i**). (**j**) Gene expression in specific endocrine cell types. Genes were selected if detected in >10% of cells in at least one of the indicated clusters and excluded if detected in >20% of enterocytes or colonocytes. Size of the circles represents percentage of expression and intensity of the circles represents aggregated expression of indicated genes partitioned by clusters.

The online version of this article includes the following source data and figure supplement(s) for figure 3:

**Source data 1.** Gene expression data .

**Figure supplement 1.** Cck, *Oc3*, and *Tph1* specify an enterochromaffin (EC) subpopulation with a dual sensory signature.

**Figure supplement 2.** Distinct molecular sensors identified in EC cells versus other enteroendocrine (EEC) cells.

with pathogen/toxin recognition (*Figure 3f*, *Figure 3—figure supplement 1*). Together, these results suggest that a dual molecular signature associated with disparate functions can be resolved in the $Cck^+/Tph1^+/Oc3^+$ cells. Additionally, numerous genes encoding enzymes/enzyme modulators (*Pcg*, *Pzp*, and *Habp2*) or proteins related with oxidation state (*Cyp2j5*, *Cyp2j6*, *Aldoc*, etc.) were highly enriched in cluster 6 cells (*Figure 3e*).

Finally, to provide a broader cellular context for the $Cck^+/Oc3^+/Tph1^+$ cells, we profiled $Neurod1^+$ EEC cells by scRNA-seq after crossing *Neurod1-Cre* mice with *Rosa26-LSL-tdTomato* mice. Among the 4397 single EEC cells retained (at threshold >500 detected genes and <10% mitochondrial transcripts per cell) from duodenum, jejunum, and colon, broad co-expression of hormone-coding transcripts was observed (*Figure 3—figure supplement 1*) as in previous reports (*Haber et al., 2017*; *Gehart et al., 2019*). To be compatible with conventional classification, clusters were annotated based on the most or second most abundant hormone-coding transcripts (*Figure 3i*, *Figure 3—figure supplement 1*). For simplicity, we assigned the diffuse clusters of $Tph1^+$ cells into either the SI EC cluster or proximal/distal colon EC clusters (based on *Hox* genes) and subdivided the $Cck^+$ cells into *Cck* dominating I cells, *Cck* and *Nts* co-expressing N cells, and *Cck* and *Gcg* co-expressing L cells, and the above described $Cck^+/Tph1^+$ cells, reasoning that *Tph1* transcripts were otherwise restricted within the EC lineage (*Figure 3—figure supplement 1*). The close relationship of I, N, and L cells evident in our dataset is in agreement with a previous study revealing a temporal progression from L to I and N cells using a real-time EEC reporter mouse model (*Gehart et al., 2019*). In addition, *Zcchc12* and *Hhex* were found to be specifically expressed in X and δ clusters, respectively, consistent with prior work (*Gehart et al., 2019*).

Among the $Neurod1^+$ EEC cells, a discrete cluster with high levels of *Cck*, *Sct*, *Tph1*, and *Ghrl*, together with *Gcg* and *Nts* transcripts, was identified and annotated as a $Cck^+/Tph1^+$ cluster (*Figure 3i*, *Figure 3—figure supplement 1*). *Oc3* and key signature genes resolved in cluster 6 were also specifically identified in this $Cck^+/Tph1^+$ population (*Figure 3j*, *Figure 3—figure supplement 1*). We thus conclude that the $Cck^+/Tph1^+$ population in the *Neurod1-tdTomato* dataset is the equivalent of the cluster 6 in the *Tph1-GFP* dataset. A previous investigation with a *Neurog3* reporter identified *Oc3* in a subset of I and N cells, but did not resolve the molecular features of these cells (*Gehart et al., 2019*). Another example where these cells may have previously been identified is a single-cell analysis of proglucagon-expressing cells, which identified a cluster expressing *Tph1*, *Cck*, *Sct*, *Ghrl*, *Gcg*, and *Nts*, along with *Casr*, *Mc4r*, and *Pzp* (*Glass et al., 2017*).

Taken together, our scRNA-seq profiling from two different genetic models, multiplex fluorescent smRNA-FISH and immunohistochemistry analysis of protein expression coordinately identified a discrete subpopulation of EEC cells preferentially located in the tip of villi in the duodenum and features a complex molecular signature, including a set of sensors associated with pathogen/toxin recognition and another set linked to nutrient sensing and homeostasis.

## Distinct molecular sensors are identified in EC cells versus other EEC cells

Having identified unique molecular sensors for various subpopulations of EC cells in the small intestine, we went on to evaluate whether these sensors are unique to EC cells by comparing them to other EECs. We focused on known and potential molecular sensors, including GPCRs, transient receptor potential channels (TRP channels), solute carrier transporters, as well as purinergic receptors and prostaglandin receptors (*Blad et al., 2012*; *Husted et al., 2017*; *Furness et al., 2013*; *Figure 3j*). In support of our previous findings, SI EC cells were preferentially enriched with *Trpa1* and *Trpm2*

along with *Cartpt* and *Ucn3* transcripts. *Casr* was enriched in the *Cck+/Tph1+* cells, whereas *Mc4r* was primarily found in the *Cck+/Tph1+* cells from the SI and additionally detected in distal colonic L cells. Consistently with the findings from the *Tph1*-bacTRAP dataset, ~99% of *Casr +*or *Mc4r+* cells in the *Cck+/Tph1+* cluster co-expressed at least one sensor gene associated with pathogen/toxin recognition (*Crp/Tlr2/Tlr5/Tril*) (*Figure 3—figure supplement 1j*), in support of our observation that *Cck+/Tph1+* cells are enriched with a dual sensory signature.

More broadly, in the EEC cells many sensors (*Cnr1*, *Asic5*, *Gper1*, and *Stra6*) demonstrated a cluster-specific expression profile, while others (*Ffar1*, *Ffar4*, *Ffar2*, and *Ffar3*) were widely distributed (*Figure 3j*). In the SI, *Cnr1* (encoding CB1), *Asic5* (encoding a bile acid sensitive ion channel (Basic)) (*Wiemuth et al., 2014*; *Wiemuth et al., 2013*), and *Gper1* (encoding G-protein-coupled estrogen receptor 1, Gpr30) were found and validated by smRNA-FISH to be enriched in the *Gip*-dominant K cell (*Moss et al., 2012*), delta cells (*Beumer et al., 2018*), and K cells, respectively (*Figure 3—figure supplement 2a–d*). In the colon, *Stra6* (encoding a retinol transporter) was exclusively identified in distal colonic L cells and mapped to *Pyy+* cells by smRNA-FISH (*Figure 3—figure supplement 2e, f*). Transcripts encoding retinol-binding proteins Rpb2 and Rpb4 have previously been identified in EECs, and have been implicated in cell differentiation processes (*Billing et al., 2019*; *Calderon et al., 2022*). *Gpbar1* (encoding a bile acid sensor Tgr5) was found in the proximal/distal colonic L cells, rather than in the EC cells. We further validated this finding by smRNA-FISH (*Figure 3—figure supplement 2g, h*) and by a similar finding from human mucosa single cells (*Figure 3—figure supplement 2i–k*). However, our study may not have detected low levels of *Gpbar1* expression. A previous study suggested that *Gpbar1* is expressed in EC cells, but not enriched compared to other cell types (*Lund et al., 2018*). In contrast to these cluster-specific sensors, the long chain fatty acid receptor *Ffar1* (*Gpr40*) was found in almost all *Cck+* cells, as well as in *Gcg+* L cells from both SI and colon. *Ffar2* (*Gpr43*) and *Ffar3* (*Gpr41/42*) encode GPCRs that recognize microbial metabolites, the former of which we widely observed in all EEC cells except K cells, whereas the latter we primarily detected in the closely related L, I, and N cells (*Figure 3j*). Lastly, almost all of these molecular sensors were confined to EEC cells, with little or no expression observed in other epithelial cell types (*Haber et al., 2017*; *Figure 3—figure supplement 2*). Therefore, we have validated the specificity of EC sensors using an independent scRNA-seq dataset together with various public datasets (*Haber et al., 2017*; *Wang et al., 2020*; *Glass et al., 2017*; *Gehart et al., 2019*) and determined the enrichment of known and newly identified chemical/nutrient sensors in various types of EEC cells.

## Two distinct clusters are identified in proximal colonic EC cells

The EC cells in the colon present distinct transcriptomic profile from their counterparts in SI. In the proximal colon, EC cells encompassed two clusters: *Iapp+/Cpb2+/Serpine1+/Sct+* cluster 11 and *Il12a+/Olfr558+/Olfr78+/Tac1+* cluster 12 (*Figure 4a*). GO analysis indicated that proximal colonic EC cells were involved in coagulation regulation (*Figure 4—figure supplement 1a*), which is supported by the selective expression of *Cpb2* and *Serpine1* (*Figure 4a*), encoding the two known plasmin inhibitors, thrombin-activatable fibrinolysis inhibitor (TAFI) and plasminogen activator inhibitor-1 (PAI-1), respectively. *Iapp* was also validated in a subset of proximal colonic EC cells (cluster 11) by smRNA-FISH, but not in distal colonic EC cells (*Figure 4—figure supplement 1b*).

Cluster 12 EC cells, on the other hand, are specialized in microorganism metabolite sensing. *Olfr558* (encoding olfactory receptor 558) and *Olfr78* (encoding olfactory receptor 78), two genes encoding GPCRs sensing short-chain fatty acids (*Fleischer et al., 2015*; *Lund et al., 2018*), were enriched in cluster 12 (*Figure 4a, b*). This is consistent with a previous report showing *Olfr78* and *Olfr558* in colonic EC cells with high expression levels of *Tac1*. *Il12a* (*Billing et al., 2019*) (encoding interleukin-12 subunit alpha) was concomitantly expressed in this cluster (*Figure 4a*) and validated by smRNA-FISH (*Figure 4c*). *Il12a* is expressed by antigen-presenting cells, such as tissue-resident macrophages and dendritic cells, to promote helper T cells differentiation in responses to microbial infection. It is possible that cluster 12 EC cells may sense pathogens and transmit hormonal signals or (a) neurotransmitter(s) to evoke immune responses in the proximal colon. Using smRNA-FISH, we further mapped *Olfr558* and *Il12a* transcripts to a separate subset of EC cells expressing *Cpb2* (*Figure 4b, c*), supporting the idea that there are subpopulations of EC cells in the proximal colon with gene transcripts associated with different physiological roles.

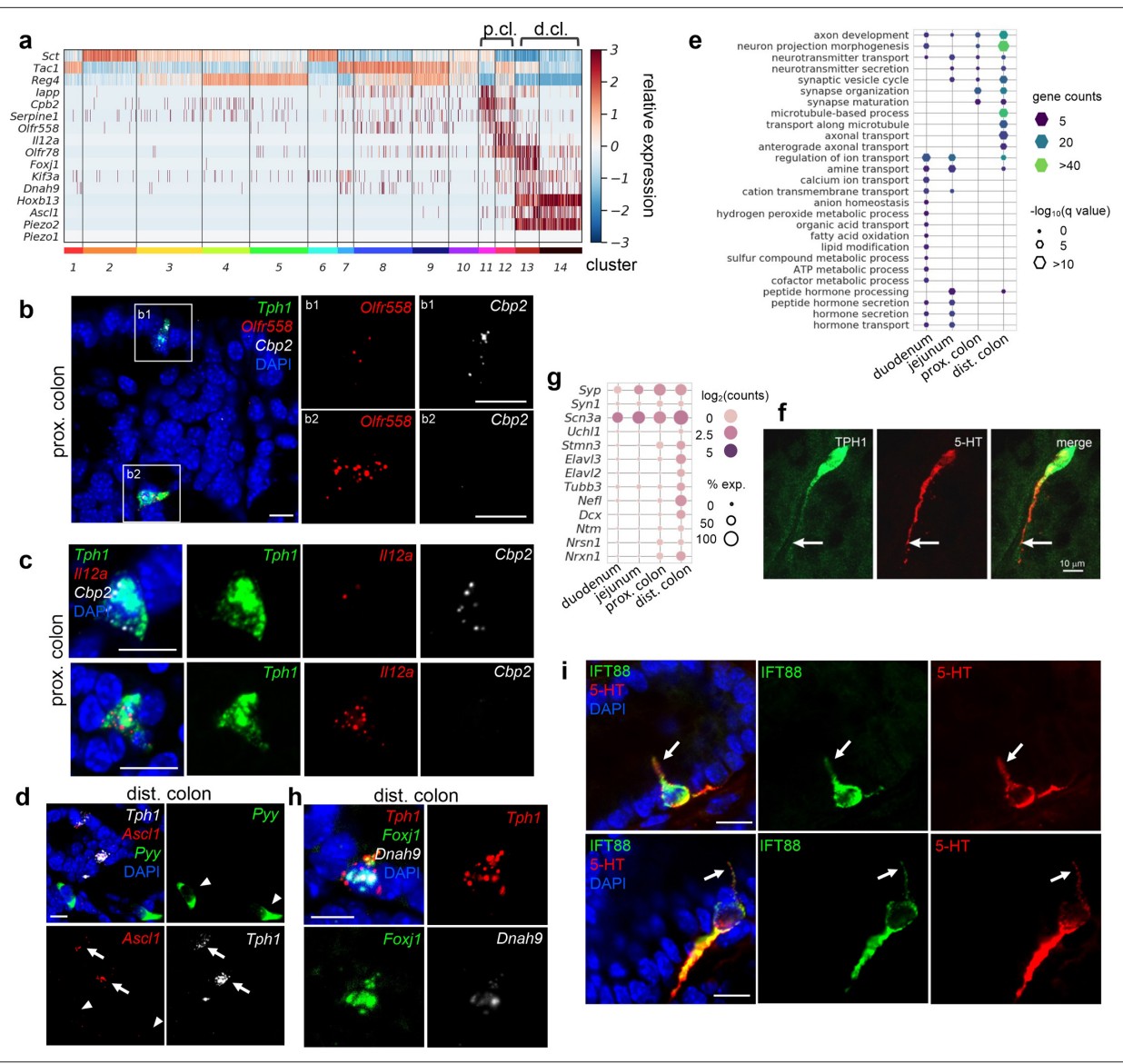

**Figure 4.** Distinct subpopulations of enterochromaffin (EC) cells are resolved in the colon. (**a**) Heatmap showing the key signature genes identified in the colonic EC cells. Relative expression (*z*-score) of indicated genes are shown across all single EC cells based on the *Tph1-bacTRAP* dataset. Color bars at the bottom represent clusters and the ones on the top represent regions. p.cl.: proximal colon; d.cl.: distal colon. (**b**) smRNA-FISH of *Tph1*, *Olfr558*, and *Cbp2* in the proximal colon. Two dashed boxes (**b1, b2**) are enlarged on the right to show two representative *Tph1⁺* cells with preferential *Cbp2* expression (**b1**) or *Olfr558* expression (**b2**). (**c**) smRNA-FISH of *Tph1*, *Il12a*, and *Cbp2* in the proximal colon. Two representative *Tph1⁺* cells with preferential *Cbp2* expression (upper) or *Il12a* expression (lower) are shown. (**d**) smRNA-FISH of *Ascl1*, *Tph1*, and *Pyy* in the distal colon. Arrows point to *Ascl1* staining in *Tph1⁺* cells. Arrowheads point to the absence of *Ascl1* staining in *Pyy⁺* cells. (**e**) GO analysis based on DE genes identified in regional EC cells. Hypothalamic neuron enriched genes were identified based on data from GEO dataset GSE74672, of which 60.7% genes were detected in gut EC cells. DE genes among the regional EC cells were determined by false discovery rate (FDR) <10⁻¹⁰ against every other region based on Wilcoxon rank sum test and corrected by Benjamini–Hochberg procedure. Size of the hexagons represents the *q*-value of enrichment after −log₁₀ transformation, and the density represents the number of genes per GO term. Accumulative hypergeometric testing was conducted for enrichment analysis and Sidak–Bonferroni correction was applied to correct for multiple testing. (**f**) IF staining against 5-hydroxytryptamine (5-HT) and GFP (representing Tph1) in the distal colon. Note the prominent axon-like extension in the distal colon EC cell. (**g**) Dot plot showing representative neuron-related genes enriched in distal colon. (**h**) smRNA-FISH of *Tph1*, *Foxj1*, and *Dnah9* in the distal colon. (**i**) IF staining against IFT88 and 5-HT in the distal colon. Data shown are two representative examples from four mice. Scale bars in panels b–d, f, i: 10 µm. Images in panels b–d, f are representative from three mice.

The online version of this article includes the following figure supplement(s) for figure 4:

**Figure supplement 1.** Distinct subpopulations of enterochromaffin (EC) cells are resolved in the colon.

## Mechanosensor *Piezo2* is enriched in neuron-like distal colonic EC cells

Next, we focused on determining the unique properties of distal colonic EC cells. *Ascl1*, encoding Mash1 (mammalian achaete-scute homolog 1), is required for early neural tube specification (*Johnson et al., 1990*; *Lo et al., 1991*; *Pang et al., 2011*; *Wapinski et al., 2013*). Unexpectedly, *Ascl1* was found in the distal colonic EC cells (*Figures 1e and 4a*; clusters 13 and 14) and was exclusively mapped to *Tph1*+ cells but not to *Pyy*+ cells in the distal colon, nor to any cell of the proximal colon or jejunum in smRNA-FISH analysis (*Figure 4d*, *Figure 4—figure supplement 1c*). Importantly, this feature was conserved in the human gut mucosa, where *ASCL1* was only detected in the *TPH1*+ cells from the rectum (*HOXB13* high) (*Figure 4—figure supplement 1*).

Expression of *Ascl1* in distal colonic EC cells suggests a possibility these cells have acquired a neuronal-like profile. To test this, we compared EC cells with neuropeptidergic hypothalamic neurons (*Romanov et al., 2017*), which produce many hormone peptides similar to those found in the gut EEC cells. Prominently, EC cells from the distal colon, in contrast to their counterparts from other GI regions, were overwhelmingly associated with GO terms 'neuron projection morphogenesis', 'axon development', and 'microtubule-based process' (*Figure 4e*, *Figure 4—figure supplement 1e*). Numerous genes indicative of neuronal identity were mutually enriched in distal colonic ECs (*Figure 4g*). Consistently, EC cells in the distal colon exhibited unique long basal processes, often extending for 50–100 µm, with 5-HT concentrated in the long processes, in contrast to the typical open-type, flask-shaped EEC cells observed in the proximal colon or SI (*Figure 4f*, *Figure 4—figure supplement 1f*; *Koo et al., 2021*). The function of these long basal processes is unknown but is unlikely to be related to synaptic transmission as there is no evidence of close apposition between the processes and neurons (*Koo et al., 2021*; *Dodds et al., 2022*). Taken together, EC cells in the distal colon demonstrated molecular and cellular characteristics reminiscent of neurons, which is distinctive from all the other gut EEC cells.

Furthermore, we identified cilium-related features in a subset of the distal colonic EC cells. In cluster 13, *Foxj1* and *Dnah9*, encoding a TF (forkhead box protein J1) and an axonemal dynein (dynein heavy chain 9, axonemal) required for cilia formation (*Lim et al., 1997*; *Yu et al., 2008*), respectively, were selectively enriched (*Figure 4a*) and validated by smRNA-FISH in the distal colonic EC cells but not in the proximal colonic EC cells or *Pyy*+ L cells (*Figure 4h*, *Figure 4—figure supplement 1g*). Genes associated with GO terms 'cilium assembly/organization' were also enriched in the distal colonic EC cells (*Figure 4—figure supplement 1a, h*). Primary cilium is a specialized cell surface projection that functions as a sensory organelle (*Singla and Reiter, 2006*), where GPCRs (e.g., rhodopsins, olfactory and taste receptors, Smo, etc.) can be located to and sense the immediate surrounding environment to activate downstream signaling(s) (*Singla and Reiter, 2006*; *Shah et al., 2009*). Consistent with the molecule features, we identified cilia by immunostaining against intraflagellar transport protein 88 (IFT88), an essential component for axonemal transportation, and found exclusive co-staining of IFT88 with 5-HT (*Figure 4i*). Gene enrichment analysis of the *Foxj1*+ EC cells also revealed concordant expression of *Olfr78* in cluster 13 (*Figure 4a*), an observation validated by smRNA-FISH (*Figure 4—figure supplement 1i*) and suggesting that a subset of distal colonic EC cells (cluster 13) represent specialized sensory cells that detect microbial products.

Most prominently, in the distal colon, *Piezo2*, encoding a mechanosensitive ion channel, was identified in almost all EC cells (*Figure 4a*). smRNA-FISH further revealed robust expression of *Piezo2* in the *Ascl1*+/*Tph1*+ cells residing in the epithelial layer of distal colon mucosa (*Figure 5a*, *Figure 5—source data 1*, *Figure 5—figure supplement 1a*, *Figure 5—figure supplement 1—source data 1*). In addition, we noticed low levels (1–5 puncta per cell) of *Piezo2* signals in the lamina propria beneath the epithelium throughout the gut which contains mainly immune cells and connective tissue cells (*Figure 5—figure supplement 1b*). In contrast to *Piezo2*, *Piezo1* transcripts were either undetectable or sparsely observed (1 or 2 puncta per cell) in epithelium and lamina propria (*Figure 5—figure supplement 1b*). *Piezo2* was not detected in the *Tph1*+ cells from the proximal colon, ileum, jejunum, and duodenum via smRNA-FISH (*Figure 5b, c*, *Figure 5—figure supplement 1b*). However, previous studies have demonstrated *Piezo2* expression in human and mouse small intestine by RT-PCR and confirmed localization in EC cells by immunohistochemistry, suggesting that *Piezo2* is expressed at levels below the detection threshold of the current study (*Wang et al., 2017*). To further evaluate the variation of *Piezo2* expression, we sorted GFP⁻ and GFP⁺ cells from various segments of the gut isolated in the *Tph1-bacTRAP* animals and quantitated *Piezo2* and other newly identified signature

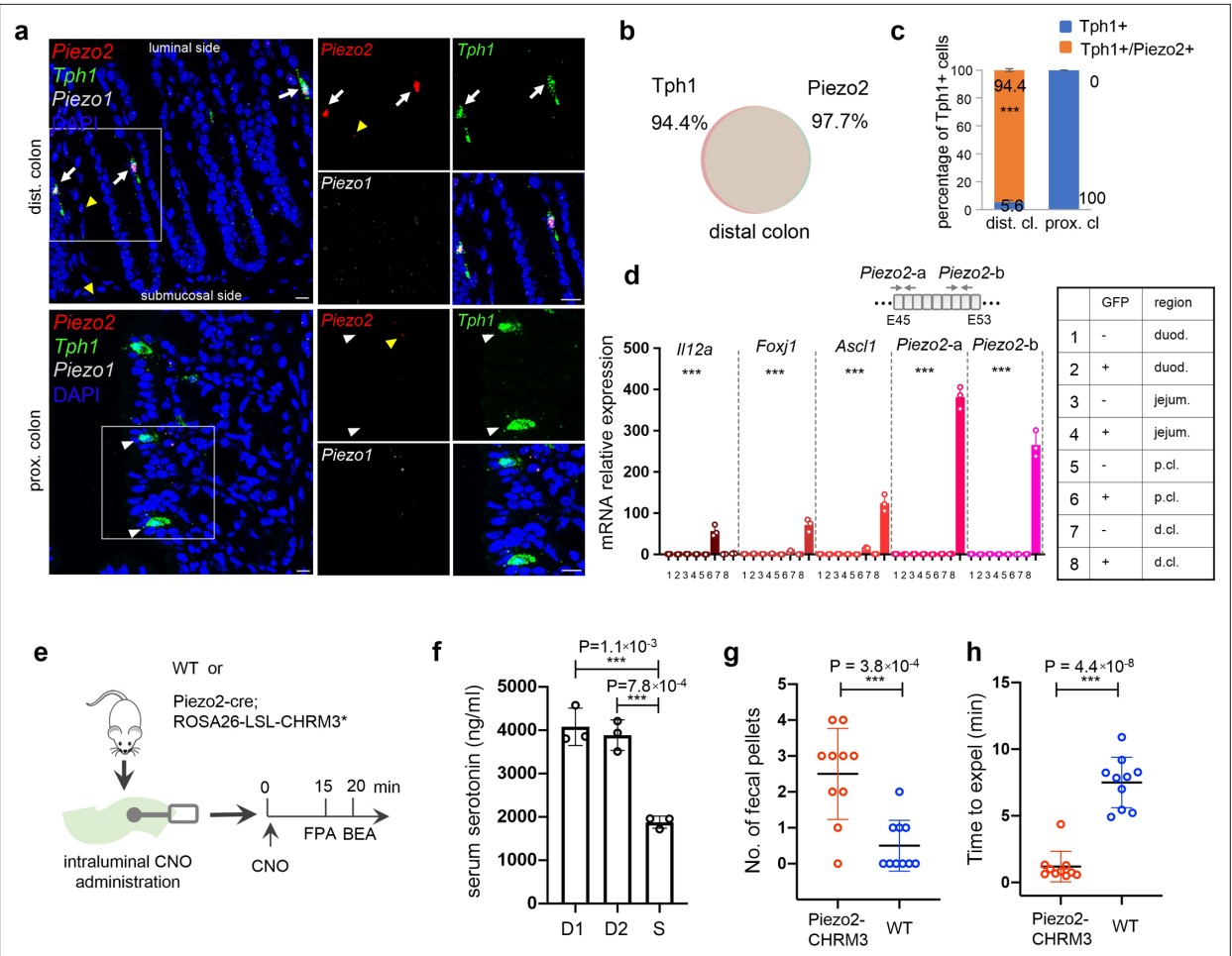

**Figure 5.** *Piezo2* is highly enriched in distal colonic enterochromaffin (EC) cells that mediate colon motility. (**a**) smRNA-FISH of *Piezo2*, *Piezo1*, and *Tph1* in distal (*upper*) and proximal colon (*lower*). Arrows point to *Piezo2/Tph1* double-positive cells in the distal colon. White arrowheads point to absence of *Piezo2* in *Tph1⁺* cells in the proximal colon. Yellow arrowheads point to sparse staining of *Piezo2* within the lamina propria. Images are representative from four mice. Scale bars: 10 μm. (**b**) Venn diagram showing the co-expression of *Tph1* and *Piezo2* transcripts in the distal colon based on smRNA-FISH. Percentages are of double-positive cells out of respective single-positive cells. A total of 274 cells were quantitated from four mice. (**c**) Quantitation of *Piezo2* and/or *Tph1* expressing cells in distal and proximal colon based on smRNA-FISH. A total of 386 cells were quantitated from four wild-type mice. ***p < 0.001; unpaired two-tailed Student's *t*-test of the double-positive fractions in distal colon versus proximal colon. (**d**) qPCR validation of regional enriched genes in EC cells. GFP⁻ and GFP⁺ cells were sorted from the duodenum, jejunum, and proximal and distal colon in the *Tph1-bacTRAP* mice. Relative gene expression was computed relative to the values in the GFP⁻ cells isolated from the duodenum (indicated as 1), after normalization by the aggregates of three house-keeping genes (*B2m*, *Gapdh*, and *Rpl13a*). Two sets of primers were used to detect *Piezo2* expression. E45: exon 45 of the *Piezo2* gene (NCBI reference sequence NM_001039485.4). ***p < 0.0001; two-way ANOVA test for both variables of GFP positivity and regions of gut, as well as the interaction of the two variables. Data are representative from three independent experiments. (**e**) Schematic of intraluminal administration of clozapine-*N*-oxide (CNO) followed by fecal pellet assay (FPA) and bead expulsion assay (BEA). FPA and BEA were conducted in the same cohorts at 15 and 20 min after CNO administration. (**f**) Serum serotonin level determined by ELISA from blood samples collected retro-orbitally 15 min after CNO administration (*n* = 3 mice per group). S: saline control, D1 (dose 1): 120 ng/kg, D2 (dose 2): 60 ng/kg. Representative data from two independent experiments are shown. Separate animal cohorts were used for serum serotonin assay versus FPA and BEA assays. (**g**) FPA from fecal pellets collected for 15 min after CNO administration. *n* = 10 in both WT and *Piezo2-CHRM3\** cohorts. Either WT or *Piezo2-cre;ROSA26-LSL-CHRM3\** (referred to as *Piezo2-CHRM3\**) mice were treated with CNO at 60 ng/kg once intraluminally. Representative data from three independent experiments. (**h**) BEA performed 20 min after CNO administration. A glass bead was inserted 2 cm into the distal colon, and the time to expel it was monitored in each animal. *n* = 10 in both cohorts. Representative data from three independent experiments.

The online version of this article includes the following source data and figure supplement(s) for figure 5:

**Source data 1.** Quantitative PCR data, fecal pellet data, bead expulsion, serotonin levels.

**Figure supplement 1.** Piezo2 is highly enriched in distal colonic enterochromaffin (EC) cells that mediate colon motility.

**Figure supplement 1—source data 1.** Serum serotonin levels, bead expulsion.

**Figure supplement 2.** Chemogenetic activation of Piezo2 cells leads to serotonin release and accelerated colon motility.

genes by qPCR analysis (*Figure 5d*, *Figure 5—figure supplement 1c and d*). A significant enrichment of *Piezo2*, up to 360-fold, was observed in the GFP+ cells sorted from the distal colon, when compared to the GFP+ cells from duodenum, jejunum, or proximal colon or to the GFP− cells from the same regions. Based on multiple lines of evidence, we conclude that *Piezo2* is preferentially enriched in neuron-like distal colonic EC cells. Furthermore, a concomitant expression of *Piezo2* with *Foxj1* and *Olfr78* was revealed by scRNA-seq (*Figure 4a*, cluster 13) and validated by smRNA-FISH (*Figure 4—figure supplement 1i*), which suggests that a subpopulation of *Piezo2+/Tph1+* cells are mechanosensory cells.

## *Piezo2+/Ascl1+/Tph1+* cells are required for normal colon motility

It is well documented that mechanical pressure or volume change within the gut lumen stimulates serotonin release from EC cells and initiates secretion and peristalsis (*Bulbring and Crema, 1958*; *Bulbring and Crema, 1959*). *Piezo2* is a mechanosensitive ion channel required in several pressure-sensing physiological systems (*Zeng et al., 2018*; *Nonomura et al., 2017*; *Woo et al., 2014*; *Marshall et al., 2020*). A previous study demonstrated that the mouse jejunum and colon express functional mechanosensitive Piezo2 channels using ex vivo assays (*Alcaino et al., 2018*). In addition, they found that mechanical stimulation evokes 5-HT release in primary colon cultures. A recent study has now shown that in an epithelial specific *Piezo2* knock-out model whole gut transit and colon transit times are slower compared to wild-type animals and, in an ex vivo colonic motility assay, small shear forces increase frequency of colonic contractions in wild-type but not knock-out animals (*Treichel et al., 2022*). We have further validated the role of Piezo2 in colon motility, with a focus on the distal colon *Piezo2+/Ascl1+/Tph1+* cells.

First, we generated a *Piezo2-CHRM3\** model by crossing *Piezo2-IRES-cre* knock-in mice (*Woo et al., 2014*) with *Cre*-dependent activating DREADD mice (*Rosa26-LSL-CHRM3\*/mCitrine*) (*Zhu et al., 2016*), such that upon administration of clozapine-*N*-oxide (CNO), *Piezo2+* cells are chemically activated in vivo (*Figure 5e*). Fifteen minutes after intracolonic administration of CNO, a robust elevation (2.1 ± 0.2 fold) of serum serotonin was observed in *Piezo2-CHRM3\** mice, but not in WT, or *Rosa26-LSL-CHRM3\** mice following the same treatment, and no effect was observed with saline administration or in untreated animals (*Figure 5f*, *Figure 5—figure supplement 2a*). To investigate serotonin release, we sorted mCitrine+/Epcam+ cells from various gut segments of the *Piezo2-CHRM3\*/mCitrine* mice and evaluated serotonin release in response to CNO in vitro (*Figure 5—figure supplement 2b–e*). Despite the low levels of mCitrine signal, we identified and sorted ~0.3% of mCitrine+/Epcam+ cells from the distal colon. In contrast, this population was largely absent from the proximal colon and duodenum, suggesting *Piezo2-cre* is primarily operational in the distal colon of the epithelial compartment. Additionally, total serotonin levels in the mCitrine+/Epcam+ cells from the distal colon were 4.7 (±0.6) fold of those in the mCitrine−/Epcam+ cells from the distal colon, proximal colon and duodenum (*Figure 5—figure supplement 2c*). Moreover, in response to CNO stimulation, serotonin release from the mCitrine+/Epcam+ cells of distal colon was elevated by 1.9 (±0.19) fold, whereas mCitrine−/Epcam+ cells failed to respond (*Figure 5—figure supplement 2d e*). Together, this data demonstrates that CNO-mediated activation of *Piezo2+* cells leads to robust serotonin release from epithelial EC cells.

We observed increased fecal pellet output from the *Piezo2-CHRM3\** mice within 15 min after CNO administration, in contrast to the WT controls (*Figure 5g*). In a bead expulsion assay (BEA), colon motility was found to be significantly accelerated, such that the time to expel an inserted bead was shortened from 7.24 (±2.0) min in wild-type controls to 1.16 (±1.17) min in the *Piezo2-CHRM3\*/mCitrine* animals after CNO treatment, while no difference was observed between wild-type controls and *Piezo2-CHRM3\*/mCitrine* animals in untreated or saline treated cohorts (*Figure 5h*, *Figure 5—figure supplement 2f*).

We noticed that *Htr4*, encoding the 5-HT4 receptor, a prokinetic 5-HT receptor when activated, was selectively expressed by epithelium in the deep crypts of distal colon (*Figure 5—figure supplement 1e*). Meanwhile, the long basal processes of EC cells filled with 5-HT always extend toward the base of the crypts, where the *Htr4* is preferentially expressed (*Figure 5—figure supplement 1*). This finding suggests close proximity of 5-HT release to its receptor.

Next, to investigate whether *Piezo2+/Ascl1+/Tph1+* ECs are required for normal colon motility, we crossed *Piezo2-IRES-cre* with *Rosa26-LSL-DTR* (diphtheria toxin receptor) (*Buch et al., 2005*) to generate *Piezo2-DTR* mice, such that upon diphtheria toxin (DT) administration, *Piezo2+* cells would

be depleted. Systemic administration of DT led to lethality in the *Piezo2-DTR* mice within 12 hr, but not in the *Rosa26-LSL-DTR* or *Piezo2-cre* mice (data not shown), likely due to the essential function of Piezo2 in respiration (*Nonomura et al., 2017*). To avoid lethality, we administrated DT intraluminally into the distal colon for 5 consecutive days and assessed distal colon motility by a 2-hr fecal pellet assay (FPA) and the BEA (*Figure 6a*, *Figure 6—source data 1*). Profound co-depletion of *Piezo2* and *Tph1* transcripts was demonstrated by smRNA-FISH in the distal colon of the *Piezo2-DTR* mice, but not in the proximal colon or in the WT mice receiving the same DT treatment (*Figure 6b, c*, *Figure 6— figure supplement 1*). Substantial loss of 5-HT$^+$ cells was further validated by immunofluorescence staining in the distal colon, but not in the proximal colon, while the general epithelial architecture was well maintained (*Figure 6b, d*, *Figure 6—figure supplement 1*, *Figure 6—figure supplement 1—source data 1*). Importantly, despite the extensive reduction of epithelial *Piezo2,* both the number and the intensity of the *Piezo2* puncta in the lamina propria of the *Piezo2-DTR* mice remained comparable to those of WT controls receiving the same DT treatment (*Figure 6—figure supplement 1e*), suggesting intraluminal administration of DT is unlikely to extensively perturb *Piezo2* expression in the lamina propria.

In a 2-hr FPA, a 42% reduction in fecal pellet output was observed in the *Piezo2*-depleted mice compared to WT animals with the same treatment regimen (*Figure 6e*). BEA demonstrated a substantial delay (36.9 ± 19.1 min) to expel an inserted bead in the *Piezo2*-depleted mice compared to the WT controls (8.2 ± 1.9 min) (*Figure 6f*), which was not observed in the *Rosa26-LSL- DTR* animals under the same treatment (*Figure 6—figure supplement 1*). Although gastric emptying was not affected in the *Piezo2-DTR* animals after DT treatment, small intestine transit (SIT) time, a measurement to assess the motility of small intestine, presented a small but statistically significant slowdown in the former group (*Figure 6g and h*). There are several possible explanations for this. Some Piezo2$^+$ cells in the small intestine could have been depleted. Alternatively, 5-HT released from Piezo2$^+$Tph1$^+$ cells in the distal colon may provide feedback to the small intestine to accelerate motility, and thus depletion of these cells would result in slower intestinal transit. Consistent with the retarded colon motility, the whole gut transit time was found to be delayed in the *Piezo2-DTR* animals (181.8 ± 17.6 min) in comparison to WT (136.7 ± 9.3 min, *Figure 6i*) under the same DT treatment.

## Epithelial Piezo2 is important for normal colon motility

To directly test whether epithelial *Piezo2* is required to maintain normal colon motility, we used *Villin-cre* to deplete Piezo2 in gut epithelial cells. Unexpectedly, 15.9% of the *Villin-cre;Piezo2$^{fl/fl}$* mice (referred to as Piezo2 CKO hereafter) died around 21–34 days after birth, affecting both males and females. By the time of humane euthanasia, the affected animals presented a 42% reduction of body weight and runt body size (*Figure 7—figure supplement 1a–c*, *Figure 7—figure supplement 1— source data 1*).

*Piezo2* depletion was observed from the isolated epithelial layer of the distal colon, as assessed by qPCR (*Figure 7a*, *Figure 7—source data 1*) and smRNA-FISH analysis (*Figure 7b*, *Figure 7— figure supplement 1i*). Anatomical and histological analysis suggested largely comparable intestine length and architecture of the gut wall with littermate controls (*Figure 7—figure supplement 1d, e*). Meanwhile, *Tph1* and *Chga* levels remained unaltered in the Piezo2 CKO animals assessed by qPCR (*Figure 7c*, *Figure 7—figure supplement 1*). Consistently, no change was observed in basal serotonin levels from either the epithelial tissue or the serum (*Figure 7—figure supplement 1g, h*). Notably, residual signals of *Piezo2* were observed in some of the *Tph1$^+$* cells of the distal colon in the Piezo2 CKO animals (*Figure 7b*), suggesting incomplete depletion. Importantly, *Piezo2* signals in the lamina propria remained largely unaltered (*Figure 7b*, *Figure 7—figure supplement 1i, j*), suggesting that only the epithelial *Piezo2* was abolished in this mouse line.

Lastly, we measured BEA and total GI transit time. A significant slowing of expulsion was revealed by BEA in the Piezo2 CKO mice (male 33.7 ± 19.0 min, female 32.3 ± 18.1 min) compared with littermate Piezo2$^{fl/fl}$ controls (male 9.25 ± 2.2 min, female 8.9 ± 1.5 min, *Figure 7d*). In addition, prolonged whole gut transit time was observed in Piezo2 CKO mice (male 182 ± 17.5 min, female 175 ± 15.6 min) compared to littermate Piezo2$^{fl/fl}$ controls (male 146 ± 11.7 min, female 143 ± 13.7 min; *Figure 7e*). To assess small intestinal transit, mice were euthanized 70 min after gavage of a fluorescent dye and the travel distance of the dye within the intestine was calculated as a percentage of total small intestinal length. No difference was observed in SIT between Piezo2 CKO mice (95.9 ± 2.6%) versus Piezo2$^{fl/}$

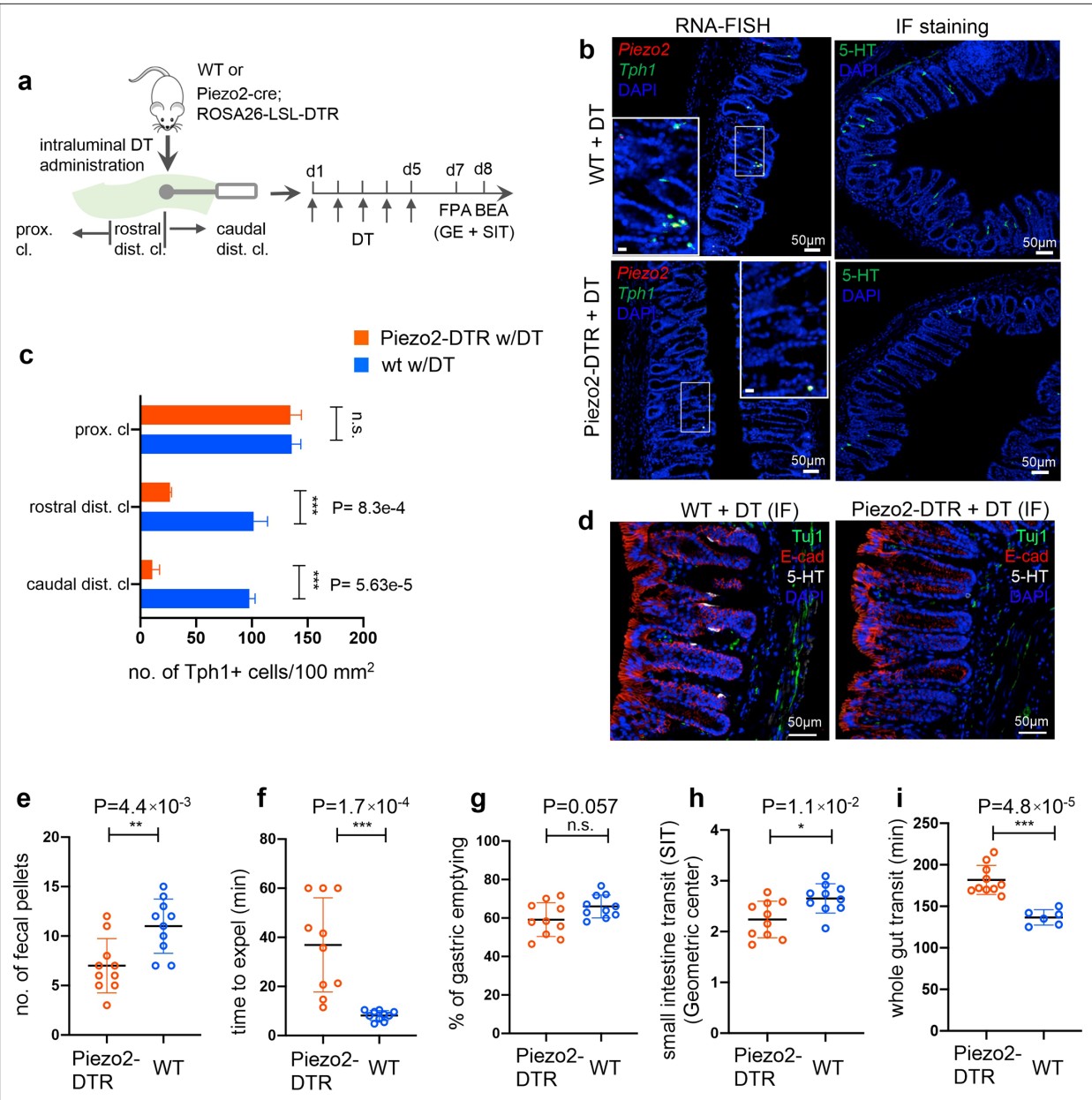

**Figure 6.** *Piezo2+/Ascl1+/Tph1+* cells are required for normal colon motility. (**a**) Schematic of intraluminal administration of diphtheria toxin (DT) followed by fecal pellet assay (FPA) and bead expulsion assay (BEA) in the same cohort. In a separate cohort, gastric emptying (GE) and small intestine transit (SIT) time assays were conducted following DT administration. Either wild-type or *Piezo2-cre;ROSA26-LSL-DTR* (referred to as *Piezo2-DTR*) mice were treated with DT at 50 μg/kg twice a day for 5 consecutive days. smRNA-FISH assay of *Piezo2/Tph1* (*left*) and IF staining for 5-hydroxytryptamine (5-HT) (*right*) from the distal colon in wild-type (*upper*) and *Piezo2-DTR* (*lower*) mice after DT administration (**b**). Dashed boxes are enlarged in the insets. Images are representative from five animals per group. Scale bars: 50 μm. (**c**) *Tph1+* cell counts based on smRNA-FISH experiments from the caudal distal colon (caudal dist. cl.), rostral distal colon (rostral dist. cl.), and the proximal colon (proximal cl.) for animals shown in (**a**) and (**b**). Five sections per animal were examined in five animals per group. (**d**) IF staining of 5-HT, E-cadherin, and Tuj1 in the distal colon of WT (left) and *Piezo2- DTR* (right) mice after DT administration. Images are representative of five different animals per group. Scale bars: 50 μm. FPA showing the number of fecal pellets collected in 2 hr (**e**), and BEA measuring the time to expel a glass bead inserted 2 cm into the distal colon (**f**). *n* = 10 per group, representative data from four independent experiments. Gastric emptying time (**g**) and SIT time (**h**) examined after 5 days of consecutive treatment of DT. Animals were orally gavaged with methylcellulose supplemented with rhodamine B dextran (10 mg/ml). Fifteen minutes after gavage, the remaining rhodamine B dextran was determined from the stomach and segments of intestine to assess upper GI motility. SIT was estimated by the position of the geometric center of the rhodamine B dextran in the small bowel. The geometric center values are distributed between 1 (minimal motility) and 10 (maximal motility). *n* = 10 in each group, representative data from three independent experiments are shown. (**i**) Whole gut transit time examined after 5 days of consecutive treatment of DT. An unabsorbable dye (carmine red) was administered by gavage and the time interval of first observance of the dye in stool was

*Figure 6 continued on next page*

*Figure 6 continued*

considered as whole gut transit time. *n* = 10 in Piezo2-DTR and 6 in wild-type control. Data are representative from two independent experiments. Error bars in panels e–i denote standard deviation of the mean; *p < 0.05, **p < 0.01, ***p < 0.001; unpaired two-tailed Student's *t*-test.

The online version of this article includes the following source data and figure supplement(s) for figure 6:

**Source data 1.** RNA-FISH cell counts and motility data.

**Figure supplement 1.** *Piezo2+/Ascl1+/Tph1+* cells required for normal colon motility.

**Figure supplement 1—source data 1.** Bead expulsion, gastric emptying and transit data.

fl controls (96.6 ± 2.2%, *Figure 7—figure supplement 1k*), in contrast to the DTR experiments, in which small intestinal transit was delayed. This could be due to the depletion of EC cells in the DTR experiments, whereas they are retained in the Villin-Cre Peizo2 KO mice. 5-HT secretion from ECs can be induced by other stimulants (even when Peizo2 is knocked out), and thus colonic 5-HT could be providing feedback to the small intestine to accelerate motility in the Villin-Cre Peizo2 KO mice. Residual Peizo2 expression in these mice could also be contributing to this effect. We then assessed

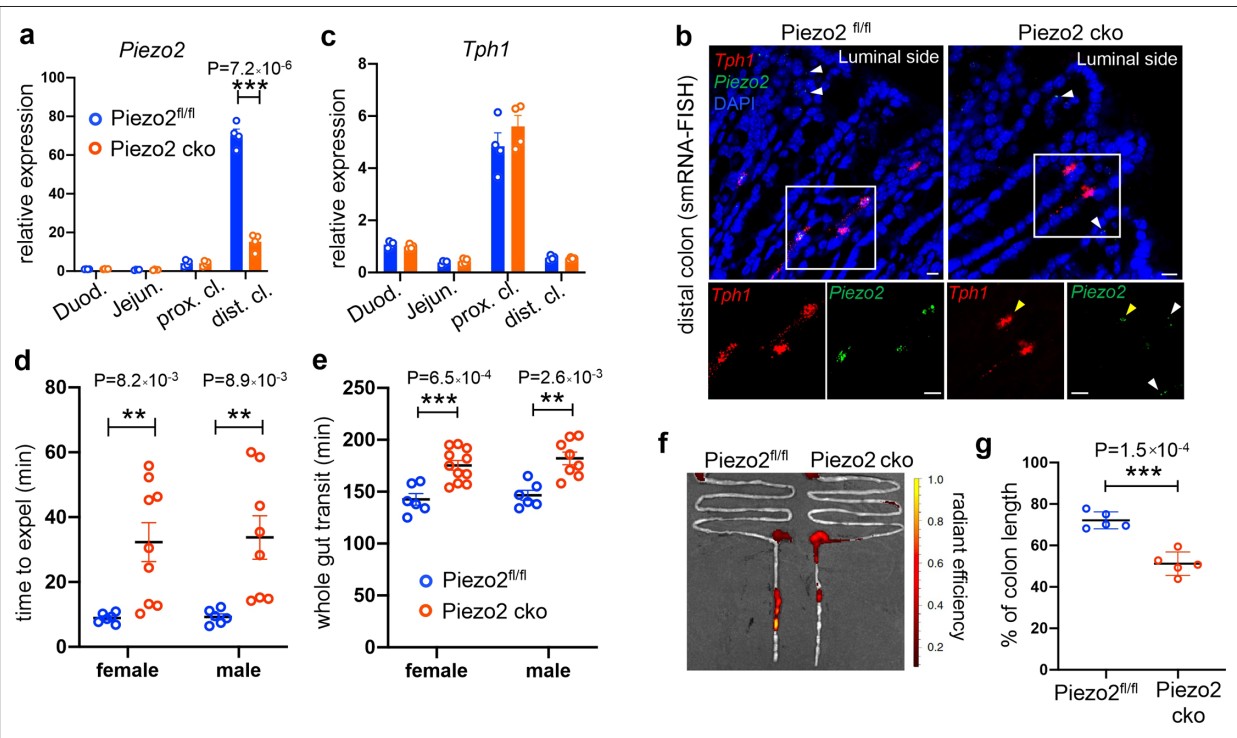

**Figure 7.** Epithelium *Piezo2* is required for efficient colon motility. (**a**) qPCR analysis of *Piezo2* depletion in *Villin-cre;Piezo2fl/fl* mice (Piezo2 cko) and *Piezo2fl/fl*. qPCR was performed on the RNA prepared from the epithelial extracts of the indicated regions of the gut. Gene expression was computed relative to the values in the *Piezo2fl/fl* duodenum, after normalization by the aggregates of three house-keeping genes (*B2m*, *Gapdh*, and *Rpl13a*). Each circle represents one animal. Data were summarized from *n* = 4 in each group. (**b**) smRNA-FISH assay of *Piezo2* and *Tph1* in either *Piezo2fl/fl* (left) or Piezo2 cko (right) mice. Dashed boxed are enlarged and presented in individual channels. Data are representative from five different animals per group. White arrowheads point to the submucosal signals of *Piezo2*. Yellow arrowheads point to the residual *Piezo2* signals in the *Tph1+* cells of the Piezo2 cko animals. Scale bars: 10 µm. (**c**) As in (**a**), qPCR analysis of *Tph1* in the Piezo2 cko and *Piezo2fl/fl* epithelium. (**d**) BEA. Each circle represents one animal. Representative data from two independent experiments. (**e**) Whole gut transit time. Each circle represents one animal. Representative data from two independent experiments. (**f**) Example fluorescent images of Piezo2 cko and *Piezo2fl/fl* intestine, 120 min after gavage of a fluorescent dye. (**g**) Summary data of fluorescent dye transit in the colon at 120 min after gavage. % of colon length = dye travel distance in colon ÷ full length of colon × 100%. Each circle represents one animal. Representative data from two independent experiments. Error bars in panels a–e, g denote standard deviation of the mean; *p < 0.05, **p < 0.01, ***p < 0.001; unpaired two-tailed Student's *t*-test.

The online version of this article includes the following source data and figure supplement(s) for figure 7:

**Source data 1.** Quantitative PCR data.

**Figure supplement 1.** Epithelium *Piezo2* is required for efficient colon motility.

**Figure supplement 1—source data 1.** Serotonin levels and transit times.

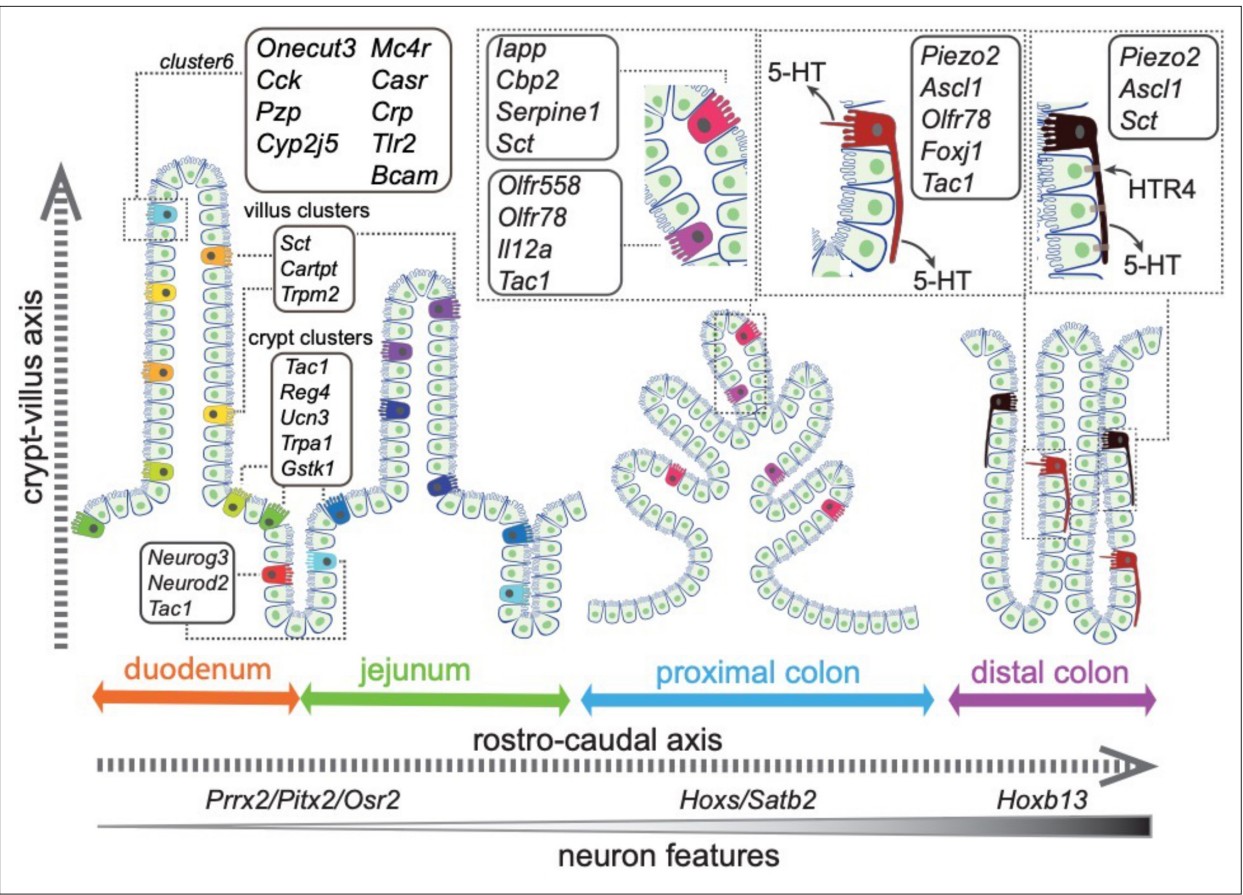

**Figure 8.** Summary of the spatial distribution of the identified EC clusters. Summary of the identified EC clusters and their respective gene signatures. Colors of the EC cells correspond to the clusters identified in this study.

fluorescent dye travel distance as a percentage of total colon length at 120 min after oral gavage, which was significantly shorter (51.2 ± 5.6%) in the Piezo2 CKO mice compared to Piezo2$^{fl/fl}$ controls (72.1 ± 4.1%, *Figure 7f, g*). Taken together, loss of Piezo2 in epithelial EC cells primarily affected colon motility.

Collectively, our in vivo gain- and loss-of-function analyses demonstrated that *Piezo2$^+$/Ascl1$^+$/Tph1$^+$* cells are required for normal colon motility. By selectively targeting a subset of EC cells expressing specific sensors – here, a mechanical sensor – our study illustrates an example to effectively untangle pleiotropic functions of a complex cell population.

## Discussion

In this study, we integrated high-throughput single-cell RNA-sequencing with spatial imaging analysis and constructed an ontological and topographical map for EC cells in mouse intestine (*Figure 8*). We resolved 14 EC subpopulations characterized by their expression of distinct chemical and mechanical sensors, TFs, subcellular structures, and explicit spatial distribution within the gut mucosa (*Figure 7h*). Together with in vivo functional validation of one subtype of EC cells, our study offers a framework to categorize complex sensory cells with defined molecular traits and led us to propose the functional identities for some of the subpopulations (*Supplementary file 1*), while others warrant future investigation due to the limited information on their roles in GI biology (e.g., *Trmp2*, *Cartpt*, and *Ucn3*).

We demonstrate that the transcriptome diversity of EC cells is closely related to their spatial distribution. The first layer of complexity is defined along the rostro-caudal axis, between the small intestine and the colon (*Figure 8*). Together with a host of TFs defining the rostro-caudal axis, we observe a gradual enrichment of a neuron-like transcriptome from small intestine to proximal and distal colon, such that *Ascl1*, encoding a key TF required for early neural fate commitment (*Johnson et al., 1990*;

*Lo et al., 1991*), is specifically expressed in the distal colonic EC cells. Concomitantly, these cells are featured with axon-like long basal processes and discrete expression of the mechanical sensor *Piezo2*. We resolved a second layer of complexity along the crypt–villus axis for the EC cells in a similar manner as described in EEC cells and enterocytes (*Billing et al., 2019*; *Moor et al., 2018*).

By combining scRNA-seq profiling from two different genetic models, smRNA-FISH and in vivo genetic ablation, our study demonstrated that *Piezo2* is preferentially expressed in distal colonic EC cells that also have neuronal expression profiles. Neuron-like features are seen in the morphology of EC in the mouse distal colon, many of these having long processes, 50–100 μm or longer, that contain 5-HT (*Kuramoto et al., 2021*). Depletion of Piezo2 from these cells caused a fourfold increase in bead expulsion time, implying that the mechanical stimulus provided by the bead caused 5-HT release and the initiation of colon propulsion. Consistent with this conclusion, in other studies in mice, 5-HT released from EC has been shown to cause colorectal propulsion (*Pustovit et al., 2021*). In addition, *Piezo2* signals were observed in the lamina propria throughout the intestine. Given the previous report that Piezo2 was detected via scRNA-seq in a subset of sensory DRG neuron innervating the colon (*Hockley et al., 2019*), it is possible that both distal colonic EC cells and the sensory DRG neurons contribute to mechanosensory sensing and motility control. Such a dual component epithelial cell–neuronal sensory machinery parallels mechanisms described in the skin (Merkel cell–neurite complexes) (*Woo et al., 2014*; *Maksimovic et al., 2014*), lung (neuroepithelial bodies) (*Nonomura et al., 2017*) and most recently in the bladder (urothelial cells-sensory neurons)[64]. However, in contrast to Piezo2 signaling in ECs which results in accelerated gut transit, Piezo2 signaling in DRG neurons appears to slow transit (*Servin-Vences et al., 2023*; *Wolfson et al., 2023*).

A cluster of *Cck+/Oc3+/Tph1+* EC cells identified in the duodenal villi was enriched with two sets of sensory molecules: enzymes/receptors associated with pathogen/toxin recognition (*Crp*, *Lyzl4*, *Bcam*, *Tril*, *Tlr2*, and *Tlr5*) and receptors associated with nutrient sensing and homeostasis (*Casr* and *Mc4r*), suggesting a that these cells react to gastric content entering the duodenum. The gut has a well-established defensive role to expel noxious chemicals and toxins by nausea and vomiting that is initiated by 5-HT release and can be effectively inhibited by 5-HT3 receptor antagonists (*Mawe and Hoffman, 2013*; *Andrews et al., 1998*). Our findings suggests that *Cck+/Oc3+/Tph1+* cells equipped with pathogen/toxin recognition receptors may play a role in this defense mechanism. This includes C-reactive protein, encoded by *Crp* selectively expressed in *Cck+/Oc3+/Tph1+* cells, which is a conserved pattern recognition molecule involved in complement-mediated cell lysis (*Black et al., 2004*; *Volanakis and Wirtz, 1979*) and lysozyme-like protein (LYZL) 4 that belongs to a family of antibacterial proteins, Toll-like receptors 2 and 5 and the Toll receptor interacting protein, TRIL. This suggests that the *Cck+/Oc3+/Tph1+* EC cells may react to pathogens both locally and through 5-HT signaling. Further study will be required to elucidate the molecular mechanism of this potentially important first line of defense. In regard to the nutrient- sensing molecules, previously, CasR and Mc4r have been reported in a subset of CCK+ I cells (*Liou et al., 2011*; *Wang et al., 2011*) and *Gcg+* L cells (*Panaro et al., 2014*), respectively, whereas *Pzp* is found to be expressed in a subset of *Gcg+* L cells (*Glass et al., 2017*). Our integrated analysis indicates that all three molecules are enriched in the specialized *Cck+/Oc3+/Tph1+* cells.

EC cells reside along the frontier between the host and a highly dynamic range of chemicals and microorganism-derived signals within the intestinal lumen that are perturbed in various diseases, and, like other EEC cells, exhibit considerable plasticity (*Beumer et al., 2020*; *Legan et al., 2022*). There are numerous reports of alterations to EC cell density in different pathophysiological states (*Mawe and Hoffman, 2013*) and, interestingly, some studies illustrating alterations to EC cell function. For example, EC cell sugar sensitivity is reduced in diet-induced obesity (*Martin et al., 2020*), and colonic ECs in patients with ulcerative colitis have altered expression of genes relating to antigen processing and presentation and to chemical sensation (*Lyu et al., 2022*). Identification of orthologous EC subtypes in humans will be an important future step toward identifying how specific EC subtypes are affected in pathophysiological states, such as celiac disease, inflammatory bowel disease, and inflammatory bowel syndrome.

## Methods

### Key resources table

See Appendix 1—key resources table.

### Materials availability statement

All sources of materials are indicated in the manuscript and/or Appendix 1—key resources table. No unique materials were created for this study.

### Animals

Mice, *Neurod1-Cre* (Jackson Labs, 028364), *Rosa26-LSL- tdTomato* (Jackson Labs, 007914), *Piezo2-IRES-cre* (Jackson Labs, 027719), *Rosa26-LSL- CHRM3\** (Jackson Labs, 026220), *Rosa26-LSL-DTR* (Jackson Labs, 007900), *Piezo$^{fl/fl}$* (Jackson Labs, 027720), *Vilin-cre* (Jackson labs, 021504), and C57BL/6, were purchased from Jackson Laboratories. Animals were housed in groups (2–5 mice/cage) in a specific pathogen-free facility provided with environmental enrichment (shelter, nesting material, etc.) and had normal immune status.

### Generation of *Tph1*-bacTRAP mice

A C57BL/6 BAC genomic clone RP23–4G4, which contains the locus of *Tph1* gene, was isolated from a RP23 mouse genomic BAC library (http://www.gensat.org/). BAC transgenic mice were produced according to published protocols (*Gong et al., 2003*). The shuttle vector (S296-1) was digested with *Asc*I and *Not*I. A 420-bp 'A box' fragment direct upstream of the ATG start codon of the *Tph1* gene was designed to be used for homologous recombination, amplified, digested with *Asc*I and *Not*I and cloned into the shuttle vector containing EGFP-RPL12. After electroporation, co-integration was identified by ampicillin resistance and verified by Southern blot using the A box sequence as the probe target. The modified BAC DNA was injected into fertilized oocytes of C57B6/J to generate the *Tph1*-bacTRAP line.

### Tissue dissociation and flow cytometry

Male mice aged 8–12 weeks were used. After the animals were sacrificed, small intestine and colon were surgically removed, rinsed with ice-cold PBS and the luminal contents flushed out with PBS using a 20-ml syringe with an 18-gauge round-tip feeding needle (Roboz Surgical Instrument, FN-7906). Duodenal and jejunal tissue was dissected between 1–5 and 13–17 cm distal of pyloric constriction. Ileum was dissected between 1 and 9 cm rostral of ileocecal junction. For the colon, the distal 4 cm tissue of descending colon was dissected as distal colon and the segment with banded lining distal to cecum was dissected as proximal colon. Dissected gut segments were inverted inside out and incubated in DMEM supplemented with 3 mM DTT (Sigma Life Science, D9779), 1 mM EDTA (Gibco, 15575-038) and 10% FBS (Gibco, 26410-079) at 37°C for 30 min with consistent rotation. The released epithelial tissue was cut into smaller pieces, triturated with a 1000-μl pipette and dissociated in a collagenase solution with 1 U of Dispase (Stem Cell Technologies, 07923), 2 mg/ml Collagenase IV (Worthington Biochemical Corporation, LS004186), and 100 U DNase I (Worthington Biochemical Corporation, LS006330) in DMEM/F12 medium for 20–30 min at 37°C with gentle mixing every 10 min. The dissociated cells were washed and filtered through a 100-μm cell strainer followed by a 40-μm cell strainer. The flow-through was spun down and filtered through another 40-μm cell strainer. The viability of the single-cell suspension was determined using trypan blue staining.

Cell pellets from the single-cell suspension were resuspended in FACS buffer (PBS, 5% FBS, and 5 mM EDTA) for staining in ice for 10 min with 7-AAD (BD Biosciences, #51-68981E). Only 7-AAD$^-$ cells were considered as viable cells. To obtain cells for scRNA-seq, GFP$^+$/7-AAD$^-$ and GFP$^-$/7-AAD$^-$ cells were collected from the *Tph1*-bacTRAP mice, while tdTomato$^+$/7-AAD$^-$ and tdTomato$^-$/7-AAD$^-$ cells are collected from the *Neurod1-tdTomato* mice. Single-cell suspensions from different segments of gut were prepared and sorted separately. Single-cell suspensions from duodenum and jejunum were prepared from a single animal, while segments of proximal colon and distal colon were pooled from two and eight animals, respectively, to acquire sufficient numbers of cells. For ileum, however, even pooling eight animals did not yield adequate number for unbiased analysis (intestinal stem cells and

TACs were disproportionally enriched in the GFP$^+$ cells, see data analysis), thus ileum was excluded from the subsequent scRNA-seq analysis.

## Library preparation and sequencing

Single-cell suspensions of freshly sorted cells were spun down to concentrate and were counted. All scRNA-seq libraries were prepared in parallel using Chromium Single Cell 3′Reagent Kits (10X Genomics; Pleasanton, CA, USA; *Tph1*-bacTRAP and small intestine of *Neurod1-tdTomato*: v2; colon of *Neurod1-tdTomato:* v3) according to the manufacturer's instructions. Generated libraries were sequenced on an Illumina HiSeq4000 instrument, followed by de-multiplexing and mapping to the mouse genome (build mm10) using CellRanger (10X Genomics, version 2.1.1). Our sequencing saturation ranged between 61.0 and 81.7%.

## Multiplex fluorescent single-molecule RNA in situ hybridization (smRNA-FISH)

To prepare tissue sections for RNA-FISH, wild-type C57BL/6 mice (8–10 weeks) were sacrificed, intestine and colon were dissected as described above, cleaned and fixed in 4% paraformaldehyde (Electron Microscopy Sciences, 15713) at 4°C overnight, and cryoprotected in 30% sucrose (Ward's Science, 57-50-1) for 24 hr before being embedded in 100% O.C.T. compound (Tissue-Tek). Cryosections were prepared at 12 µm. Single-molecule RNA-FISH was performed using the RNAscope. Multiplex Fluorescent Detection Kit v2 (323100, Advanced Cell Diagnostics) according to the manufacturer's instructions using TSA with Cy3, Cy5, and/or Fluorescein (Perkin Elmer NEL760001KT). All sections were counterstained with DAPI (1:1000; Invitrogen D1306). All co-staining was performed on tissue samples from at least three mice. Images were acquired on Zeiss LSM780 confocal microscope. Subsequent processing of images was performed in Fiji (*Schindelin et al., 2012*; *Rueden et al., 2017*), including channel merging, pseudocoloring, maximum intensity projection, and brightness adjustment.

## Quantification of smRNA-FISH

To quantitate crypt–villus distribution of cells, 9 × 9 tile images (1024-pixel × 1024-pixel) of an entire cross section of duodenum were acquired on Zeiss LSM780 and stitched in ZEN software. For automated analysis, images were preprocessed in Fiji (2.0.0) by maximum intensity projection, background subtraction and contrast enhancement, and the processed images were output as binary images for each channel. A custom CellProfiler pipeline was built to quantitate co-staining of two or three channels. Briefly, regions of interest (ROIs) were manually drawn on the DAPI channel in order to count cells separately in the crypts versus the villi. For each set of images, objects were identified independent in each channel, the signal of which was expanded by a 5-pixel diameter circle. If the circular objects identified from different channels overlapped, the cells were identified as double- or triple-positive. For each experiment, the pipeline was automated through images collected from at least 3 sections per animal in three animals. The number of double-positive cells (e.g., *Tph1$^+$/Cartpt$^+$*) was divided by single-positive cells (e.g., *Tph1$^+$*) in each ROI and summarized across three animals to calculate mean and SEM statistics were performed using an unpaired Student's *t*-test to compare enrichment in the crypts versus in the villi. To quantitate co-expression of two or three feature genes, the same pipeline was employed without demarcation of crypts versus villi.

## Quantification of Piezo2 depletion via smRNA-FISH

Sections were prepared from distal colon of either Piezo2-DTR or WT mice after intraluminal treatment of DT for 5 days. A custom CellProfiler pipeline was built to quantitate staining of *Tph1* and *Piezo2* channels. For each set of images, puncta (objects) were identified in the *Tph1* or *Piezo2* channel independently. When objects identified from the two channels overlapped, the puncta were identified as double positive; otherwise, the puncta were identified as single positive. The intensity of the *Piezo2* signals was quantitated by the size of puncta. Data were summarized from four pairs of animals with two tiled images from each animal. smRNA-FISH probes are listed in Appendix 1—key resources table.

## qPCR analysis

Cells were dissociated from the stripped epithelial layer in the duodenum, jejunum, and proximal and distal colon of the *Tph1*-bacTRAP animals. Total RNA was prepared from the sorted regional GFP$^-$ and GFP$^+$ cells using the RNeasy Micro Kit (QIAGEN). First-strand cDNA was synthetized from 100 ng RNA using the SuperScript III First-Strand Synthesis System (Thermo Fisher, 18080051). Quantitative PCR was performed using FastStart Universal SYBR Green Master Mix (Sigma, 4913850001). The aggregates of three housekeeping genes (*B2m*, *Gapdh*, and *Rpl13a*) were used to compute deltaCt. Relative gene expression was calculated by normalization against GFP$^-$ cells extracted from the duodenum. Oligonucleotide sequences are listed in Appendix 1—key resources table.

## Tissue processing for immunohistochemistry

Duodenal (descending duodenum), jejunal (segment distal to root of mesentery), proximal colon (segment with banded lining distal to cecum), and distal colon (straight descending colon) tissue were isolated from three 2-month wild-type C57Bl/6 male mice and three 8-month-old female *Tph1*-bacTRAP mice. The segments were opened along the mesenteric attachment, pinned onto balsa wood with the mucosal side facing up, and then fixed overnight in fixative (2% (vol/vol) formaldehyde plus 0.2% (vol/vol) picric acid in 0.1 M sodium phosphate buffer, pH 7.2) at 4°C. Following fixation, tissue was washed three times in dimethyl sulfoxide, 10 min each, followed by three washes in PBS, 10 min each. Tissue was stored in PBS containing 0.1% (vol/vol) sodium azide until ready for use. Tissue was prepared for processing by placing segments in 50% (vol/vol) PBS-sucrose-azide and 50% (vol/vol) O.C.T. mixture for 24 hr before being embedded in 100% O.C.T. compound.

## Immunohistochemistry and image analysis

Sections (12 μm) were cut and allowed to dry at room temperature for 1 hr on microscope slides (SuperFrostPlus; Menzel-Glaser; Thermo Fisher, Victoria, Australia). Sections were next incubated with 10% (vol/vol) normal horse serum prepared in PBS containing 1% (vol/vol) Triton X-100 for 30 min, followed by overnight incubation at 4°C with primary antibodies (see Appendix 1—key resources table). Sections were washed three times in PBS and incubated with secondary antibodies (see Appendix 1—key resources table) for 1 hr at room temperature. Following three washes with distilled water, sections were stained with Hoechst 33258 solution (10 μg/ml in distilled water) for 5 min to allow visualization of nuclei. Slides were washed with distilled water and coverslipped using non-fluorescent mounting medium (Dako, Carpinteria, CA, USA). Slides were allowed to dry overnight at room temperature after which they were imaged at ×40 magnification using the AxioImager microscope (Zeiss, Sydney, Australia), or with the LSM800 (Zeiss) at ×20 magnification. Immunoreactive cells were quantified by counting approximately 100 cells from each region of the gut for each of the three animals.

## Serotonin measurements

Blood samples were collected retro-orbitally from indicated animal cohorts. Animals subjected to blood collection were not used for in vivo motility assays. Serum was separated after coagulation at room temperature for 60 min, meaning that platelets could be contributing to the serotonin levels measured. Serotonin levels were detected in sera by ELISA according to the manufacturer's instructions (Eagle Biosciences). To examine tissue serotonin levels from epithelial tissue, epithelium was extracted by incubation in DMEM supplemented with 3 mM DTT (Sigma Life Science, D9779), 1 mM EDTA (Gibco, 15575-038) and 10% FBS (Gibco, 26410-079) at 37°C for 30 min with consistent rotation. Dissociated epithelial cells were lysed with standard buffer provided in ELISA kits. Cleared supernatant was used for ELISA. For the serotonin secretion assay, ~1000 sorted cells were equilibrated in standard buffer (+0.1% wt/vol ascorbic acid) for 30 min after wash with PBS. Cells were then incubated with standard buffer supplemented with indicated drugs or vehicle control, as well as 0.1% ascorbic acid for 15 min. Supernatant and cell lysates were prepared to measure secreted serotonin and cell lysate serotonin separately, the sum of which is calculated as total serotonin level. Secreted serotonin was considered as supernatant serotonin ÷ total serotonin × 100%.

## CNO and DT administration

CNO (Tocris, 4936) was dissolved in sterile 0.9% saline (Quality Biological, 114-055-101). On the day of experiment, the animals were anesthetized with isoflurane (5%, 1 l/min). CNO at indicated doses or saline control were administrated intracolonically using a 1-ml syringe with a 22-gauge round-tip feeding needle (round tip diameter 1.25 mm, length 36 mm; Roboz Surgical Instrument, FN-7920). Initially, two different doses (60 ng/kg, 120 ng/kg) of CNO were tested. As they were equally potent to induce serum serotonin elevation, the lesser dose (60 ng/kg in 50 µl) was chosen for the rest of experiments. DT (Sigma, D0564) was reconstituted in sterile water and administrated to anesthetized animals in the same manner as CNO, at 50 µg/kg in 50 µl twice a day for 5 consecutive days. Different regimens of DT administration were tested initially to obtain the maximal depletion of *Piezo2+* cells in the distal colon.

## In vivo motility assays

Animals were assigned to a random number before functional assays and data were collected in a blinded manner.

## Fecal pellet assay

The night before the assay, all animals (8–10 weeks, male) were housed singly in regular cages with wire mesh bottoms and bedding underneath with free access to food and water. After overnight acclimatization, each animal was placed into new wire mesh bottom cages. Fecal pellets were collected over 2 hr and counted for each animal. For the cohorts receiving CNO treatment, since increased fecal pellets were already observed within the first 15 min after CNO administration in the *Piezo2-CHRM3\** mice, we have shortened the collection time window to 15 min for this experiment.

## Bead expulsion assay

All animals (8–10 weeks male or as indicated in the manuscript) were fasted for 2 hr before assay. Briefly, the animals were anesthetized with isoflurane (5%, 1 l/min). A glass bead (2 mm in diameter, Sigma 1040140500) was placed into the colon using a disposable feeding needle with silicone tip (Fisherbrand, 01-208-89) to a distance of 2 cm from the anal verge. Mice were returned to their individual cages and allow to come back to full consciousness. Time required to expel the glass bead was monitored and recorded in each animal. The experiment was terminated at 60 min after mice became fully conscious. The mice failed to expel glass beads within this time window were reported as 60 min.

## Gastric emptying and SIT analysis

All animals (8–10 weeks, male) were fasted overnight in cages that lacked bedding. Water was withdrawn 3 hr before the experiment. Mice were orally gavaged with 100 µl sterile solution of 10 mg/ml rhodamine B dextran (Sigma R9379) in 2% methylcellulose (Sigma, M7027) through a 20-gauge round-tip feeding needle (Roboz Surgical Instrument, FN-7903). Animals were scarified 15 min after gavage; the stomach, small intestine, cecum, and colon were collected PBS. The small intestine was divided into 10 segments of equal length, and the colon (used to obtain total recovered rhodamine B fluorescence) was divided in half. Each piece of tissue was homogenized in PBS and centrifuged ($2000 \times g$) to obtain a clear supernatant. Rhodamine fluorescence was measured in 250 µl aliquots of the supernatant (Tecan Trading, Infinite M200). Gastric empty rate was calculated as [(total recovered fluorescence − fluorescence remaining in the stomach) ÷ (total recovered fluorescence)] × 100%. Small intestinal transit was estimated by the position of the geometric center of the rhodamine B dextran in the small bowel (*Miller et al., 1981*). For each segment of the small intestine (1–10), the geometric center (*a*) was calculated as follows: *a* = (fluorescence in each segment × number of the segment) ÷ (total fluorescence recovered in the small intestine). The total geometric center is $\Sigma$ (*a* of each segment). Total geometric center values are distributed between 1 (minimal motility) and 10 (maximal motility).

## Total gastrointestinal transit time analysis

Male animals between 8 and 16 weeks were used, or as indicated in the manuscript when both female and male animals were examined. The night before the experiment, animals were transferred to individual housing with free access to water only. On the day of experiment, animals had free access to

food and water for 1 hr. A solution of 6% carmine red (300 μl, Sigma, C1022) was prepared using 0.5% methylcellulose (Sigma, M7027) and was administered by gavage through a 21-gauge round-tip feeding needle (Roboz Surgical Instrument, FN-7903). 90 min after gavage, fecal pellets were monitored for the presence of carmine red. Total GI transit time was calculated from the time of administration to the first observance of carmine red in stool.

### In vivo transit analysis

Male animals between 8 and 16 weeks were used for this assay. Animals were fasted overnight and then had free access to food and water for 1 hr. 100 μl Gastrosense 750 (Perkin Elmer) prepared in PBS was administered to the stomach via gavage. To assess partial intestinal transit, the GI tract (stomach to terminal colon) was removed at indicated time. Fluorescence images were obtained of the GI tract using an IVIS Lumina II In Vivo Imaging System (Perkin Elmer).

### scRNA-seq data processing

#### Gene expression matrices

Gene expression matrices were generated using the CellRanger software (10X Genomics). Sample data were aggregated using CellRanger (cellranger -aggr) and resulting data were processed further in Python (version 3.6.2) or scanpy (version 2.1.4) (**Wolf et al., 2018**).

#### Quality control

The following quality control steps were performed: (1) non-coding gene and genes expressed in fewer than 10 cells were not considered; (2) cells that expressed fewer than 500 genes were excluded from further analysis (suggesting a low quality of cells); (3) cells in which >10% of unique molecular identifiers (UMIs) were derived from the mitochondrial genome were removed.

#### Cell doublet removal

Since we kept our capture rate low (~2000 cells per sample) during library preparation, the cell doublet rate was low as expected. We removed potential cell doublets based on: (1) the presence of gene signatures from two different cell classes, such as epithelial markers and immune cell markers; (2) the observation of a second peak of total UMIs distribution in comparison to the cells from the same class.

#### Normalization

Given the presence of stochastic zeros in scRNA-seq data and the wide distribution of total UMIs per cells in our dataset, we used a pool-based size factor to deconvolute cell-specific size factor for each single-cell library for normalization purpose. Briefly, a coarse cell clustering was first performed using hierarchical clustering based on the top 50 components of principle component analysis (PCA) for the entire expression matrix. Summation of UMI counts across cells in each resolving cluster was computed as pooled-based size factor and repeated to generate a linear system for all single cells. A weighted least-squares approach is applied to solve the linear system and to deconvolute a cell-base size factor for all single cells, as implemented in Scran (version 2.1.6) (**Lun et al., 2016**). The UMI counts were then normalized by a cell-specific size factor and transformed as $\log_2$(normalized counts + 1). For simplicity, the transformed normalized counts are presented as $\log_2$(counts).

#### Clustering and spatial visualization

Linear dimensionality reduction was performed on the aggregated dataset using PCA. The top PCs were chosen based on elbow plots, where the percentage variance explained by each PC was plotted, and the number of principal components was chosen as a substantial drop was observed in the proportion of variance explained. Typically, 20–30 PCs were chosen based on our dataset, and were visualized using t-distributed stochastic neighbor embedding (t-SNE) (**Van der Maaten and Hinton, 2008**). Graph-based clustering was performed for community detection. Briefly, a k-nearest neighbors (kNN) graph was built based on the Euclidian distance of the single cells in the PCA space, the edges between the detected community were weighted using Jaccard similarity, and the Louvain method was applied to optimize the modularity of the communities. Cluster numbers were chosen based on Bayesian information criterion and biological considerations. Cluster resolution (parameter of k),

t-SNE perplexity and the number of PCs used for clustering and visualization were adjusted based on the total cell number of the dataset (related to *Figure 1d, g, h*; *Figure 1—figure supplement 1c–e, g*; *Figure 3i*).

## Identification of EC and non-EC cells

In the initial scRNA-seq profiling from the *Tph1*- bacTRAP mice, we obtained 4729 signal cells including GFP⁻ (2412) and GFP⁺ (2317) cells. Clustering analysis coupled with spatial visualization in t-SNE space, as described above, indicated ~23% of the GFP⁺ cells clustered together with the GFP⁻ cells. To annotate GFP⁻ cells and GFP⁺ clustered with GFP⁻ cells, we obtained gene sets from GSE92332 (*Haber et al., 2017*), which include gene sets associated with each major cell types in the intestinal epithelium that have been identified, including intestinal stem cells, TACs, immature enterocytes, mature enterocytes, tuft cells, goblet cells, and EEC cells. We calculated module scores for each cell cluster identified in our dataset by computing the average expression levels of each cell type gene set subtracted by the aggregated expression of all detected genes in our dataset. Cell types were assigned based on the highest module scores across all cell types as described above. Furthermore, we applied *z*-score transformation to the cell type-enriched gene set and validate the cell type assignment based on their respective *z*-score enrichment. The same approaches were employed to identify the non-EC cell types in the second scRNA-seq profiling from the *Tph1*-bacTRAP mice, thus annotated intestinal stem cells, TACs, immature enterocytes, mature enterocytes and colonocytes. The T lymphocytes and mast cells were identified based on the top differentially expressed genes in the respective clusters (related to *Figure 1d*, *Figure 1—figure supplement 1d*).

## Elimination of ileal GFP⁺ cells

Previous studies have indicated a significant decline of 5-HT⁺ cells in the distal small intestine (*Reynaud et al., 2016*). Consistent with such observation, in the initial scRNA-seq profiling of the single epithelial cells from *Tph1*-bacTRAP mice, we obtained only 225 GFP⁺ single-cell libraries from the ileal epithelial cells collected from eight *Tph1*-bacTRAP mice, which was significantly lower than those from other regions of the gut (0.1% GFP positivity compared to ~0.3–0.5% positivity in other regions). Additionally, the total number of epithelial cells obtained from the ileum in each mouse was only 20% of those from duodenum, contributing to the lower number of GFP⁺ cells obtained from the ileum. Upon further computational analysis, as much as 72% (162) of these GFP⁺ cells clustered with non-EC cells, in particular, 49% (111) were identified to have marker genes for stem cells or TACs, indicating that a majority of these cells are not fully committed EC cells. We thus eliminated ileum from further analysis.

## Identification of differentially expressed genes in cell populations

To identify genes expressed at significantly higher level in one cluster or region than the other clusters or regions, we used the Wilcoxon rank-sum test, which is non-parametric and does not assume normality. Correction for multiple testing was performed using the Benjamini–Hochberg procedure to control the FDR. We ran differential expression tests between each pair of clusters (regions) for all possible pairwise comparisons, as implemented in Scran (version 2.1.6). For a given cluster (region), the DE genes were filtered using the maximum FDR *q*-values across all pairwise comparisons. Genes that are known to be associated with dissociation process were not considered to be differentially expressed genes (*van den Brink et al., 2017*). For *Figures 1l, 2f, and 4e*, DE genes were obtained using maximum FDR $<10^{-10}$.

## Identification of signature genes for clusters

To identify maximally specific genes for each EC cell clusters, we performed pairwise differential gene expression analysis as described above between each pair of clusters for all possible pairwise comparisons. For a given cluster, putative signature genes were ordered based on FDR (FDR $<10^{-10}$, the smallest FDR was ranked on top). The final signature genes lists were obtained by calculating the rank product for selected genes in all pairwise comparisons. Rank product statistic (*Heskes et al., 2014*) was used to determine p values for each marker gene and was adjusted by the Benjamini–Hochberg procedure to correct multiple hypothesis testing. To ensure the signature genes are enriched in EC

cells but not other cell types in gut epithelium, the candidate genes were evaluated against both GFP$^-$ and non-EC cells identified in our own datasets and against the GSE92332 dataset (*Haber et al., 2017*).

## GO term enrichment analysis

Differentially expressed genes identified by regions (*Figure 1h*) or by clusters (*Figure 2h*, *Figure 2— figure supplement 1*) were selected by FDR <10$^{-10}$ and subjected to enrichment analysis by accumulative hypergeometric test followed by Sidak–Bonferroni correction as implemented in Metascape (*Zhou et al., 2019*). The resulting GO terms were selected by *q*-values that are presented by the size of the hexagons. The number of DE genes identified in the GO terms is represented by heatmap.

## Comparison with public datasets

We compared neuropeptidergic hypothalamic neurons with EC cells based on two considerations: (1) hypothalamic neurons produce many hormone peptides similar to gut EEC cells, and (2) colonic EC cells are enriched with many neuronal signatures. We extracted expression matrices and metadata from GSE74672 (*Romanov et al., 2017*). Cell types were catalogued as indicated by metadata. Data were processed in the same pipeline as described for the *Tph1*-bacTRAP or *Neurod1*$^+$ dataset. DE gene analysis was performed to identify neuron-enriched genes against both microglia and oligodendrocytes (FDR <10$^{-10}$). The resulting gene set was used to compare EC cells originated from different segments of the gut. To identify orthogonal signature genes from human gut mucosa, we cross-compared the *Neurod1*$^+$ dataset with GSE125970[21], where human gut mucosal cells were isolated from biopsy samples in ileum, colon and rectum. Since no selection has been applied to dissociated human epithelial cells, <1% of captured cells are EEC cells. We identified 63 L cells and 34 EC cells in total, which were determined as *Pyy*$^+$, *Gcg*$^+$, or *Pyy*$^+$/*Gcg*$^+$ for the former, and *Tph1*$^+$ for the latter. Data were processed in the same manner as *Tph1*-bacTRAP or *Neurod1*$^+$ dataset.

## Statistics

In addition to the statistic tests described in the data analysis section, the following tests were performed:

For *Figures 1i and 2j* and *Figure 2—figure supplement 1g, h*; A two-sample Kolmogorov–Smirnov test was performed to test whether two underlying probability distributions are the same. Implemented by scipy.stats.ks_2samp in python.

For *Figure 2d*: An unpaired two-tailed Student's *t*-test was employed to test whether the positive fractions identified in the villi are significantly different from the ones observed in the crypts.

For *Figure 2i*: Hypergeometric testing for enrichment of indicated hormones in the *Cck*$^+$/*Tph1*$^+$ population versus the Tph1$^+$ population was performed and implemented by scipy.stats.hypergeom in Python.

## ARRIVE guidelines

This study adheres to ARRIVE 2.0 guidelines. Essential details are included within the manuscript and figures.

## Acknowledgements

We thank Ardem Patapoutian for conducive discussions. Stefan Aigner is thanked for the major contributions to assembling the paper and associated files. We acknowledge the help of Jesus Olvera and Cody Fine with FACS and Elsa Molina for assistance with confocal microscopy. Confocal microscopy was undertaken at the Biological Optical Microscopy Platform, University of Melbourne. Funding: This work is partially funded by a grant from the Takeda-Sanford Innovation Alliance. Sequencing was conducted at the Institute for Genomic Medicine (IGM) Genomics Core at UC San Diego which is supported by P30CA023100. MP was supported by National Research Service Award NRSA grant F32HL143978. The funders had no role in study design, data collection, and interpretation, or the decision to submit the work for publication.

# Additional information

## Competing interests

Jie Huang: At the time of these studies JH was an employee of Takeda Pharmaceutical Company International Inc, and held stock and/or stock options in Takeda. Jill Wykosky: At the time of these studies JW was an employee of Takeda Pharmaceutical Company International Inc, and held stock and/or stock options in Takeda. The other authors declare that no competing interests exist.

## Funding

| Funder | Grant reference number | Author |
|---|---|---|
| Takeda Pharmaceuticals U.S.A. | | Gene W Yeo |
| National Institutes of Health | F32HL143978 | Mark Perelis |

The funders had no role in study design, data collection, and interpretation, or the decision to submit the work for publication.

## Author contributions

Yan Song, Investigation, Writing – original draft; Linda J Fothergill, Investigation, Writing – original draft, Writing – review and editing; Kari S Lee, Brandon Y Liu, Shanti Diwakarla, Jie Huang, Investigation; Ada Koo, Formal analysis, Investigation; Mark Perelis, Brid Callaghan, Investigation, Methodology; Jill Wykosky, Conceptualization, Funding acquisition, Writing – original draft; John B Furness, Conceptualization, Formal analysis, Investigation, Writing – original draft, Project administration, Writing – review and editing; Gene W Yeo, Conceptualization, Funding acquisition, Investigation, Writing – original draft, Project administration, Writing – review and editing

## Author ORCIDs

John B Furness ⬥ https://orcid.org/0000-0002-0219-3438
Gene W Yeo ⬥ https://orcid.org/0000-0002-0799-6037

## Ethics

This study was performed in strict accordance with the recommendations in the Guide for the Care and Use of Laboratory Animals of the National Institutes of Health. All animal procedures performed at the University of California San Diego were conducted with approval by the Institutional Animal Care and Use Committee (Approval S12099). All animal procedures at the University of Melbourne were conducted according to the National Health and Medical Research Council of Australia guidelines and were approved by the University of Melbourne Animal Experimentation Ethics Committee (Approval 10180).

Reviewer #2 (Public review): https://doi.org/10.7554/eLife.90596.3.sa1
Author response https://doi.org/10.7554/eLife.90596.3.sa2

# Additional files

## Supplementary files

MDAR checklist

Supplementary file 1. Summary of EC clusters and their potential physiological roles.

## Data availability

All data generated or analyzed during this study are included in the manuscript and supporting files; source data files have been provided for Figures 1–3, 5–7. Raw fastq files and Cell Ranger processed digital gene expression matrixes (DGE) have been deposited at NIH's NCBI BioProject under PRJNA623218.

The following dataset was generated:

| Author(s) | Year | Dataset title | Dataset URL | Database and Identifier |
|---|---|---|---|---|
| Song Y, Fothergill LJ, Lee KS, Liu BY, Koo A, Perelis M, Diwakarla S, Callaghan B, Huang J, Wykosky J, Furness JB, Yeo GW | 2025 | Stratification of enterochromaffin cells as chemosensor and mechanosensors by single cell analysis | https://www.ncbi.nlm.nih.gov/bioproject/?term=PRJNA623218 | NCBI BioProject, PRJNA623218 |

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

# Appendix 1

## Appendix 1—key resources table

| Reagent type (species) or resource | Designation | Source or reference | Identifiers | Additional information |
|---|---|---|---|---|
| Strain (*Mus musculus*) C57Bl6 | Neurod1-Cre | Jackson Labs | JAX 028364 | |
| Strain (*M. musculus*) C57Bl6 | Rosa26-LSL- tdTomato | Jackson Labs | JAX 007914 | |
| Strain (*M. musculus*) C57Bl6 | Piezo2-IRES-cre | Jackson Labs | JAX 027719 | |
| Strain (*M. musculus*) C57Bl6 | Rosa26-LSL- CHRM3 | Jackson Labs | JAX 026220 | |
| Strain (*M. musculus*) C57Bl6 | Rosa26-LSL-DTR | Jackson Labs | JAX 007900 | |
| Strain (*M. musculus*) C57Bl6 | Piezo$^{fl/fl}$ | Jackson Labs | JAX 027720 | |
| Strain (*M. musculus*) C57Bl6 | Vilin-cre | Jackson Labs | JAX 021504 | |
| Antibody | anti-5-HT (Goat; polyclonal) | ImmunoStar, | Cat# 20079, RRID:AB_572262 | IF(1:10,000) |
| Antibody | anti-5-HT (Rabbit; polyclonal) | ImmunoStar, | Cat# 20080, RRID:AB_572263 | IF(1:1000) |
| Antibody | anti-Secretin (Rabbit; polyclonal) | Phoenix Pharmaceuticals | Cat# H-067–04, RRID:AB_2650428 | IF(1:2000) |
| Antibody | anti-CCK/ gastrin (Mouse monoclonal) | Gift | Code: 28.2 RRID:AB_2650429 | IF(1:2000) |
| Antibody | anti-CCK/ gastrin (Rabbit; polyclonal) | Gift | Code: 8007/5 RRID:AB_2884979 | IF(1:8000) |
| Antibody | anti-GIP (Rabbit; polyclonal) | Santa Cruz | Cat# SC23554, RRID:AB_2247481 | IF(1:500) |
| Antibody | anti-Ghrelin (Rabbit; polyclonal) | Santa Cruz | Code MF1601, RRID:Ab_2767291 | IF(1:10,000) |
| Antibody | anti-GFP (Rabbit; polyclonal) | Abcam | Cat# AB13970, RRID:AB_300798 | IF(1:2000) |
| Antibody | anti-Oxyntomodulin (Mouse monoclonal) | Anshlabs | Cat# 323AO010, RRID:AB_2934156 | IF(1:2000) |
| Antibody | anti-Substance P (Rabbit; polyclonal) | Gift | Cat# SK1, RRID:AB_2814842 | IF(1:1600) |
| Antibody | anti-Neurotensin (Goat; polyclonal) | Santa Cruz | Cat# SC-7592, RRID:AB_650372 | IF(1:50) |
| Antibody | anti-IFT88 (Rabbit; polyclonal) | Proteintech | Cat# 13967-1AP, RRID:AB_2121979 | IF(1:500) |
| Antibody | anti-Tuj1 (Rabbit; polyclonal) | Abcam | Cat# Ab18207, RRID:AB_444319 | IF(1:1000) |
| Antibody | anti-e-cadherin (Rat; polyclonal) | Abcam | Cat# Ab11512, RRID:AB_298118 | IF(1:1000) |
| Antibody | anti-Dclk1 (Rabbit; polyclonal) | Abcam | Cat# Ab31704, RRID:AB_873537 | IF(1:1000) |
| Antibody | anti-Mouse IgG Alexa 488 (Rabbit; polyclonal) | Molecular Probes | Cat# 21202, RRID:AB_141607 | IF(1:500) |

*Appendix 1 Continued on next page*

*Appendix 1 Continued*

| Reagent type (species) or resource | Designation | Source or reference | Identifiers | Additional information |
|---|---|---|---|---|
| Antibody | anti-Chicken IgG Alexa 488 (Goat; polyclonal) | Jackson ImmunoResearch | Cat# 103–545155, RRID:AB_2337390 | IF(1:500) |
| Antibody | anti-Rabbit IgG Alexa 568 (Donkey; polyclonal) | Molecular Probes | Cat# A10042, RRID:AB_2534017 | IF(1:800) |
| Antibody | anti-Sheep IgG Alexa 488 (Donkey; polyclonal) | Molecular Probes | Cat# A11015, RRID:AB_141362 | IF(1:500) |
| Antibody | anti-Goat IgG Alexa 555 (Donkey; polyclonal) | Molecular Probes | Cat# A21432, RRID:AB_141788 | IF(1:800) |
| Sequence-based reagent | Mm-Tph1 | ACDBio/Bio-Techne | 318701 C2 | Probe used for smRNA-FISH |
| Sequence-based reagent | Mm-Piezo2 | ACDBio/Bio-Techne | 400191 C3 | Probe used for smRNA-FISH |
| Sequence-based reagent | Mm-Gpx3 | ACDBio/Bio-Techne | 400191 C3 | Probe used for smRNA-FISH |
| Sequence-based reagent | Mm-Cartpt | ACDBio/Bio-Techne | 432001 | Probe used for smRNA-FISH |
| Sequence-based reagent | Mm-Stra6 | ACDBio/Bio-Techne | 450321 | Probe used for smRNA-FISH |
| Sequence-based reagent | Mm-Cck | ACDBio/Bio-Techne | 402271 C3 | Probe used for smRNA-FISH |
| Sequence-based reagent | Mm-Foxj1 | ACDBio/Bio-Techne | 317091 | Probe used for smRNA-FISH |
| Sequence-based reagent | Mm-Dnah9 | ACDBio/Bio-Techne | 556771 C3 | Probe used for smRNA-FISH |
| Sequence-based reagent | Mm-Mc4r | ACDBio/Bio-Techne | 402741 | Probe used for smRNA-FISH |
| Sequence-based reagent | Mm-Casr | ACDBio/Bio-Techne | 423451 | Probe used for smRNA-FISH |
| Sequence-based reagent | Mm-Asic5 | ACDBio/Bio-Techne | 588601 C3 | Probe used for smRNA-FISH |
| Sequence-based reagent | Mm-Pyy | ACDBio/Bio-Techne | 420681 C3 | Probe used for smRNA-FISH |
| Sequence-based reagent | Mm-Gpbar1 | ACDBio/Bio-Techne | 318451 | Probe used for smRNA-FISH |
| Sequence-based reagent | Mm-Trpm2 | ACDBio/Bio-Techne | 313291 | Probe used for smRNA-FISH |
| Sequence-based reagent | Mm-Ascl1 | ACDBio/Bio-Techne | 313291 | Probe used for smRNA-FISH |
| Sequence-based reagent | Mm-Olfr558 | ACDBio/Bio-Techne | 316131 C2 | Probe used for smRNA-FISH |
| Sequence-based reagent | Mm-Htr4 | ACDBio/Bio-Techne | 408241 | Probe used for smRNA-FISH |
| Sequence-based reagent | Mm-Piezo1 | ACDBio/Bio-Techne | 500511 | Probe used for smRNA-FISH |
| Sequence-based reagent | Mm-Onecut3 | ACDBio/Bio-Techne | 583241 | Probe used for smRNA-FISH |
| Sequence-based reagent | Mm-Crp | ACDBio/Bio-Techne | 583251 | Probe used for smRNA-FISH |

*Appendix 1 Continued on next page*

*Appendix 1 Continued*

| Reagent type (species) or resource | Designation | Source or reference | Identifiers | Additional information |
|---|---|---|---|---|
| Sequence-based reagent | Mm-Cpb2 | ACDBio/Bio-Techne | 583261 | Probe used for smRNA-FISH |
| Sequence-based reagent | Mm-Tlr2 | ACDBio/Bio-Techne | 317521 | Probe used for smRNA-FISH |
| Sequence-based reagent | Mm-Tlr5 | ACDBio/Bio-Techne | 451601 | Probe used for smRNA-FISH |
| Sequence-based reagent | Mm-F5 | ACDBio/Bio-Techne | 502411 | Probe used for smRNA-FISH |
| Sequence-based reagent | Mm-Gip | ACDBio/Bio-Techne | 451601 | Probe used for smRNA-FISH |
| Sequence-based reagent | Mm-Cnr1 | ACDBio/Bio-Techne | 420721 C2 | Probe used for smRNA-FISH |
| Sequence-based reagent | Mm-Ucn3 | ACDBio/Bio-Techne | 464861 | Probe used for smRNA-FISH |
| Sequence-based reagent | Mm-Il12a | ACDBio/Bio-Techne | 414881 | Probe used for smRNA-FISH |
| Sequence-based reagent | Mm-Iapp | ACDBio/Bio-Techne | 512571 C2 | Probe used for smRNA-FISH |
| Sequence-based reagent | Mm-Olfr78 | ACDBio/Bio-Techne | 436601 | Probe used for smRNA-FISH |
| Sequence-based reagent | Foxj1-F | IDT DNA | N/A | qPCR primer 5' AGCCCAGAAGACTGGGAACT 3' |
| Sequence-based reagent | Foxj1-R | IDT DNA | N/A | qPCR primer 5' AATCCTTGGGCTTGAGGGAAC 3' |
| Sequence-based reagent | Ascl1-F | IDT DNA | N/A | qPCR primer 5' GAATGGACTTTGGAAGCAGG ATG 3' |
| Sequence-based reagent | Ascl1-R | IDT DNA | N/A | qPCR primer 5' TGCCCCTGTAGGTTGGCTG 3' |
| Sequence-based reagent | Piezo2-a-F | IDT DNA | N/A | qPCR primer 5' GCACTCTACCTCAGGAAGACTG 3' |
| Sequence-based reagent | Piezo2-a-R | IDT DNA | N/A | qPCR primer 5' CAAAGCTGTGCCACCAGGTTCT 3' |
| Sequence-based reagent | Piezo2-b-F | IDT DNA | N/A | qPCR primer 5' TCAAACACGCCAGTGACAAT 3' |
| Sequence-based reagent | Piezo2-b-R | IDT DNA | N/A | qPCR primer 5' TGTCTCTGAACAAAATGATG GTGA 3' |
| Sequence-based reagent | Trpa1-F | IDT DNA | N/A | qPCR primer 5' GAGGATTGCTATGCAGGTGGA 3' |
| Sequence-based reagent | Trpa1-R | IDT DNA | N/A | qPCR primer 5' CGTGCCTGGGTCTATTTGGA 3' |
| Sequence-based reagent | Chga-F | IDT DNA | N/A | qPCR primer 5' CCAAGGTGATGAAGTGCGTC 3' |
| Sequence-based reagent | Chga-R | IDT DNA | N/A | qPCR primer 5' GGTGTCGCAGGATAGAGAGGA 3' |
| Sequence-based reagent | Tph1-F | IDT DNA | N/A | qPCR primer 5' TGTTGACTGCGACATCAGCCGA 3' |
| Sequence-based reagent | Tph1-R | IDT DNA | N/A | qPCR primer 5' GGAAACCAAGGGACAGTCTC CA 3' |

*Appendix 1 Continued on next page*

*Appendix 1 Continued*

| Reagent type (species) or resource | Designation | Source or reference | Identifiers | Additional information |
|---|---|---|---|---|
| Sequence-based reagent | Trpm2-F | IDT DNA | N/A | qPCR primer 5' AAGGATGTGGCTCTCACAGAC 3' |
| Sequence-based reagent | Trpm2-R | IDT DNA | N/A | qPCR primer 5' CGGGAACCCATACTCGACC 3' |
| Sequence-based reagent | B2m-F | IDT DNA | N/A | qPCR primer 5' CACTGAATTCACCCCCACTGA 3' |
| Sequence-based reagent | B2m-R | IDT DNA | N/A | qPCR primer 5' TGTCTCGATCCCAGTAGACGG 3' |
| Sequence-based reagent | Rpl13a-F | IDT DNA | N/A | qPCR primer 5' AGCAGATCTTGAGGTTACGGA 3' |
| Sequence-based reagent | Rpl13a-R | IDT DNA | N/A | qPCR primer 5' GGAGTCCGTTGGTCTTGAGG 3' |
| Sequence-based reagent | Gapdh-F | IDT DNA | N/A | qPCR primer 5' CTGGAGAAACCTGCCAAGTATG 3' |
| Sequence-based reagent | Gapdh-R | IDT DNA | N/A | qPCR primer 5' AGAGTGGGAGTTGCTGTTGAAG 3' |
| Sequence-based reagent | Cartpt-F | IDT DNA | N/A | qPCR primer 5' AAGAAGTACGGCCAAGTCCC 3' |
| Sequence-based reagent | Cartpt-R | IDT DNA | N/A | qPCR primer 5' CAGTCACACAGCTTCCCGAT 3' |
| Sequence-based reagent | Ucn3-F | IDT DNA | N/A | qPCR primer 5' AAGGCCAAGAATTTGCGAGC 3' |
| Sequence-based reagent | Ucn3-R | IDT DNA | N/A | qPCR primer 5' TGTCTTGATGTGCCACCCTC 3' |
| Sequence-based reagent | Il12a-F | IDT DNA | N/A | qPCR primer 5' CCACTGGAACTACACAAGAACG 3' |
| Sequence-based reagent | Il12a-R | IDT DNA | N/A | qPCR primer 5' ATGCTACCAAGGCACAGGGT 3' |
| Software, algorithm | ImageJ | NIH | RRID:SCR_003070 | https://imagej.net/ij/ |
| Software, algorithm | Zen Blue | Zeiss | RRID:SCR_013672 | |
| Software, algorithm | Prism | Graphpad | RRID:SCR_002798 | https://www.graphpad.com/ |
| Software, algorithm | CellRanger | 10 X Genomics | RRID:SCR_023221 | v. 4.0.0 |
| Software, algorithm | Python | Python Software Foundation | RRID:SCR_008394 | v. 3.6.2 |
| Software, algorithm | Scanpy | Scanpy development team | RRID:SCR_018139 | v. 2.1.4 |
| Commercial assay or kit | RNeasy Micro Kit | Qiagen | 74104 | |
| Commercial assay or kit | SuperScript III First-Strand Synthesis System | ThermoFisher | 18080051 | |
| Commercial assay or kit | FastStart Universal SYBR Green Master Mix | Sigma | 4913850001 | |
| Commercial assay or kit | Serotonin ELISA Assay Kit | Eagle Biosciences | EA602 | |
| Other | DAPI stain | Invitrogen | D1306 | (1 µg/mL) |

