## [Editor Report · eLife Assessment]

This **important** study presents a transcriptomic analysis of enterochromaffin cells in the intestine. The evidence supporting the authors' claims is **solid**, although the functional analysis is focused on the Piezo2-expressing subset in the colon. The work will be of interest to biologists working on intestinal mucosal biology.

---

## [Referee Report · Reviewer #2 (Public review)]

Summary:

The authors investigated the expression profile of enterochromaffine (EC) cells after creating a new tryptophan hydroxylase 1 (Tph1) GFP-reporter mouse using scRNAseq and confirmative RNAscope analysis. They distinguish 14 clusters of Tph1+ cells found along the gut axis. The manuscript focuses on two of these, (i) a multihormonal cell type shown to express markers of pathogen/toxin and nutrient detection in the proximal small intestine, and (ii) on a EC-cluster in the distal colon, which expresses Piezo2, rendering these cells mechanosensitive. In- and ex- vivo data explore the role of the mechanosensitive EC population for intestinal/colonic transit, using chemogenetic activation, diptheria-toxin receptor dependent cell ablation and conditional gut epithelial specific Piezo2 knock-out. Whilst some of these data are confirmative of previous reports - Piezo2 has been implicated in mechanosensitive serotonin release previously, as referred to by the authors - the data are solid and emphasize the importance of mechanosensitive serotonin release for colonic propulsion. The transcriptomic data will guide future research.

Strengths:

The transcriptomic data, whilst confirmative, is more granular than previous data sets. Employing new tools to establish a role of mechanosensitive EC cells for colonic and thus total intestinal transit.

Weaknesses:

(1) The proposed villus/crypt distribution of the14 cell types is not verified adequately. The RNAscope and immunohistochemistry samples presented do not allow assessment if this interpretation is correct - spatial transcriptomics, now approaching single cell resolution, likely will help to verify this claim.

(2) The physiological function and/or functionality of most of the transcriptomically enriched gene products has not been assessed. Whilst a role for Piezo2 expressing cells for colonic transit is convincingly demonstrated the nature of the mechanical stimulus or the stimulus-secretion coupling downstream of Piezo2 activation is not clear.

Comments on revisions: I am happy with the manuscript as is.

---

## [Author Response]

The following is the authors’ response to the original reviews.

**Reviewer #1 (Public Review):**
The authors have performed extensive work generating reporter mice and performing single-cell analysis combined with in situ hybridization to arrive at 14 clusters of enterochromaffin (EC) cells. Then, they focus on Piezo channel expression in distal EC cells and find that these channels might play a role in regulating colonic motility. Overall, this is an informative study that comprehensively classifies EC cells in different regions of the small and large intestine. From a functional point of view, however, the authors seem to ignore the fact that the expression of Piezo-2-IRES-Cre is broad, which would raise concerns regarding their physiological conclusions.The authors may wish to consider the following specific points:It is surprising that the number of ileal EC cells is less than that of the distal colon, and it would be interesting to know whether the authors can comment about ileal EC cells. It is unclear why ileal ECs were not included in the study, even though they are mentioned in the diagram (Fig. 2c).

We have discussed the rationale for excluding ileal ECs in the methods section under “Elimination of ileal GFP+ cells”. In our initial scRNA-seq experiment, our yield of epithelial cells and GFP positive cells was low, and a large proportion of these cells appeared to not have fully committed to the EC lineage. Also to note, we have previously seen fewer ECs in the distal ileum than upper small intestine and colon (PMID: 26803512). Given the low yield, and some uncertainty regarding the nature of the ileal EC population sorted by our methods, we considered that data from ileal ECs may not be an accurate representation of ileal EC cell diversity. Thus, we did not use ileal ECs in our second scRNA-seq experiment.

Based on their analysis, there are 10 EC cell clusters in SI while there are only 4 clusters in the colon. The authors should comment on whether this is reflective of lesser diversity among colonic ECs or due to the smaller number of colonic ECs collected.

The 4 clusters identified in the colon are consistent with previous a previous publication (Glass et al., *Mol. Metab.* 2017, PMID: 29031728), supporting the idea that these clusters are representative of the major clusters of colonic ECs. Nonetheless, we anticipate that with greater sample sizes (in any region) further resolution of subtypes could be resolved.

The authors previously described that distal colonic EC cells exhibit various morphologies (Kuramoto et al., 2021). Do Ascl1(+) EC cells particularly co-localize with EC cells with long basal processes? Also, to validate the RNA seq data, the authors might show co-localization between Piezo2/Ascl1/Tph1 in distal EC cells. It would be interesting to see whether Ascl1-CreER (which is available in Jax) specifically labels distal colonic EC cells as this could provide a good genetic tool to specifically manipulate distal colonic EC cells.

We have shown co-localization between Piezo2/Ascl1/Tph1 in Supplementary Figure 6a. Unfortunately we did not study cell morphology in the Ascl1 smRNA-FISH experiments as these used thin cryosections, whereas morphological assessment of EC processes is best performed with thick (>60 µm) sections. It would be interesting if neuronal-like expression profiles correlate with neuronal-like morphology, which could be addressed in future studies with spatial transcriptomics.

The authors used Piezo2-IRES-Cre mice, whose expression is rather broad. They might examine the distribution of Chrm3-mCitrine in the intestine (IF/IHC would be straightforward). And if the expression is in other cell types (which is most likely the case), they should justify that the observed phenotype derives from Piezo2-expressing EC cells. Alternatively, they could use Piezo2-Cre;ePetFlp (or Vil-Flp);Chrm3 to specifically express DREADD receptors in distal colonic EC cells. Also, what does 5HT release look like in jejunal EC cells in Piezo-CHRM3 mice?

Unfortunately we no longer have access to the animals to do these experiments.

For the same reasons as above, DTR experiments may also be non-specific. For example, based on the IF staining (Fig. 6b,d), there seems to be a loss of Tph1+ cells in the proximal colon of Piezo2-DTR mice, so the effects of the Piezo2-DTR likely extend beyond the distal colon.

Figures 6b and d show distal colon, not proximal colon. Our Tph1^+^ cell counts indicate there was no loss of Tph1 cells in the proximal colon following intraluminal administrations of DT.

It is unclear why the localized loss of Piezo2 in Piezo2-DTR mice alters small intestinal transit (Fig. 6g,h). The authors should discuss the functional differences observed between Piezo2-DTR (intraluminal app) and Vil1Piezo2 KO mice i.e., small intestinal transit, 5HT release, etc. Are these differences due to the residual Piezo2 expression in Piezo2 KO mice? In this context, the authors may want to discuss their findings in the context of recent papers, such as those from the Patapoutian and Ginty groups.

We have made the following amendment to speculate on the reason for delayed small intestinal transit in the DTR experiments:

“There are a several possible explanations for this. Some Piezo2+ cells in the small intestine could have been depleted. Alternatively, 5-HT released from Piezo2+Tph1+ cells in the distal colon may provide feedback to the small intestine to accelerate motility, and thus depletion of these cells would result in slower intestinal transit.”

We have also added a comment speculating on why we did not see similar slowing of small intestinal transit in the Villlin-Cre Piezo2 KO:

“No difference was observed in small intestine transit… in contrast to the DTR experiments, in which small intestinal transit was delayed. This could be due to the depletion of EC cells in the DTR experiments, whereas they are retained in the Villin-Cre Piezo2 KO mice. 5-HT secretion from ECs can be induced by other stimulants (even when Piezo2 is knocked out), and thus colonic 5-HT could be providing feedback to the small intestine to accelerate motility in the Villin-Cre Piezo2 KO mice. Residual Piezo2 expression in these mice could also be contributing to this effect.”

We have added a comment on neural Piezo2 in the discussion:

“However, in contrast to Piezo2 signalling in ECs which results in accelerated gut transit, Piezo2 signalling in DRG neurons appears to slow transit (refs: Wolfson et al., Cell 2023; PMID: 37541195; Servin-Venves et al., Cell 2023, PMID: 37541196).”

**Reviewer #2 (Public Review):**
Summary:The authors investigated the expression profile of enterochromaffin (EC) cells after creating a new tryptophan hydroxylase 1 (Tph1) GFP-reporter mouse using scRNAseq and confirmative RNAscope analysis. They distinguish 14 clusters of Tph1+ cells found along the gut axis. The manuscript focuses on two of these, (i) a multihormonal cell type shown to express markers of pathogen/toxin and nutrient detection in the proximal small intestine, and (ii) on a EC-cluster in the distal colon, which expresses Piezo2, rendering these cells mechanosensitive. In- and ex- vivo data explore the role of the mechanosensitive EC population for intestinal/colonic transit, using chemogenetic activation, diptheria-toxin receptor dependent cell ablation and conditional gut epithelial specific Piezo2 knock-out. Whilst some of these data are confirmative of previous reports - Piezo2 has been implicated in mechanosensitive serotonin release previously, as referred to by the authors - the data are solid and emphasize the importance of mechanosensitive serotonin release for colonic propulsion. The transcriptomic data will guide future research.Strengths:The transcriptomic data, whilst confirmative, is more granular than previous data sets. Employing new tools to establish a role of mechanosensitive EC cells for colonic and thus total intestinal transit.Weaknesses:(1) The proposed villus/crypt distribution of the 14 cell types is not verified adequately. The RNAscope and immunohistochemistry samples presented do not allow assessment of whether this interpretation is correct - spatial transcriptomics, now approaching single-cell resolution, would be likely to help verify this claim.

Spatial transcriptomics would be excellent in validating the spatial distribution of the EC cell types in future studies. In our work, although the villus/crypt cluster annotations are assumptions (based on the differential expression of Neurog3, Tac1, and Sct, which is well supported by the literature), we have validated the spatial segregation of key markers. We quantified the crypt/villus location of Cartpt, Ucn3, and Trpm2 overlap with Tph1 (Figure 2d), Oc3, Cck, and Tph1 (Figure 3d), and TK/5-HT (Supplementary Fig 2d). This work supports our predictions on the spatial distribution of these clusters.

(2) The physiological function and/or functionality of most of the transcriptomically enriched gene products has not been assessed. Whilst a role for Piezo2 expressing cells for colonic transit is convincingly demonstrated, the nature of the mechanical stimulus or the stimulus-secretion coupling downstream of Piezo2 activation is not clear.

While we have not investigated the mechanical forces involved in activating Piezo2, we can at least say that physiological mechanical stimulation activates Piezo2, as we measured fecal pellet output in the DTR experiments.

**Reviewer #2 (Recommendations For The Authors):**
(1) Please state (even more) clearly if/that the apparently GFP+/Tph1+ cells which clustered with the GFP- cells (Suppl. Fig1d/e) were excluded from the subsequent analysis. The detectable Chg-a/b expression in the GFP- cells in Suppl. Fig1f seems to suggest that these (if they have been included in the GFP- group here) are genuine ECs. How do these cells relate to the non-EC cells in Fig1d, which seem to lack Tph1 expression? And given the information in the methods, what %age of these cells derived from the ileum?

To clarify, data shown in Suppl. Fig 1d/e/f was from our first single cell profiling experiment whereas our subsequent clustering analysis utilizes data from a second (independent) single cell profiling experiment (e.g. Fig1d).

In the first profiling experiment, 23% of GFP^+^ cells clustered with GFP^-^ cells, and for the purposes of Suppl. Figures 1d/e/f, we called these “non-ECs”. In the second profiling experiment (e.g. shown in Fig 1d) we performed a more detailed cluster analysis focusing on only GFP^+^ cells. In this second experiment, 19% of GFP^+^ cells were identified as “non-EC cells” based on the presence of markers for stem cells, transit amplifying cells (TACs), immature enterocytes, mature enterocytes, colonocytes, T lymphocytes and mucosal mast cells (see Fig 1d and Suppl. Fig 1g). Similar to the first profiling dataset, many of the GFP^+^ “non-EC cells” in the second dataset express Tph1, Chga, and Chgb, generally at lower levels than the “EC cells” (Suppl. Fig1i). It is possible that the stem cell and transit amplifying cell clusters are cells that are differentiating into EC cells. However, given that they have not fully committed to the lineage yet, we do not consider it appropriate to classify them as “EC cells”. With regards to the other “non-EC” clusters, we do not think that the expression of EC cell marker genes (Tph1, Chga, and Chgb) is evidence enough to call them genuine “EC cells” given the concurrent expression of markers of other lineages (e.g. enterocyte and mast cell markers Suppl. Fig 1g). The expression of Tph1 in murine mast cells is known, however the expression in enterocytes is unexpected and could be a result of imperfect/incomplete differentiation. Since the ileum was not included in the second profiling experiment we do not think the GFP^+^ “non-EC cells” are an artifact from the ileum.

We have made some adjustments in the first section of the results to clarify some thoughts on this matter:

“It is possible that some GFP is expressed in cells that have not yet fully committed to the EC lineage, or that there is some expression in cells outside this lineage, for example, in mast cells. Given the small sample size, we did not further investigate these cells in this dataset. In Supplementary Figures 1 d and f we refer to the GFP^+^ cells that clustered with the GFP^-^ cells as “non-EC cells”.”

“It is possible that the stem cell and transit amplifying cell clusters include cells that are in the process of differentiating into EC cells. However, given that they have not fully committed to the lineage, we do not consider it appropriate to classify them as “EC cells” for the purposes of analyzing EC cell types in this study.”

(2) The authors state: "Notably, OSR2 and HOXB13 were restricted to the ileum and rectum respectively in humans (Fig. 1f)." - the statement regarding OSR2 seems too strong, given that only the ileal part of the human small intestine was examined and that there is a small signal in the proximal colon in Figure 1f.

Thanks, we have made the following amendment:

"Notably, OSR2 and HOXB13 were preferentially enriched in the ileum and rectum respectively in these human samples (Fig. 1f)."

(3) Please clarify Suppl Fig2g/h labelling as villus and crypt enrichment ("...enrichment in villus clusters (g) or crypt clusters (h)."), when enrichment for some genes in cluster 4 is shown in both g and h. Why was duodenal cluster 6 excluded from this subset of data?

We suspect (although have not proven) that cluster 4 is at a later stage in maturation/migration than cluster, as indicated by a somewhat ‘middle ground’ level of Sct expression, and generally being ‘in between’ the villus clusters and cluster 5 in expression levels of differentially expressed genes shown in Suppl Fig 2g/h. We have added the following comment to the figure legend to clarify this. We have not included cluster 6 as it is transcriptionally quite distinct from the other clusters:

“Note that cluster 4 shares some features in common with crypt and villus clusters and may represent cells at an intermediate stage of development.”

(4) "Using smRNA-FISH, we further mapped Olfr558 and Il12a transcripts to a separate subset of EC cells expressing Cpb2 (Fig. 4b,c), confirming the presence of two subpopulations of EC cells associated with different physiological roles in the proximal colon." - Claiming populations with different physiological functionality seems a strong statement given the relatively weak Cpb2 signals observed and that mRNA detection necessarily is a transcriptomic time limited snap-shot. Please reformulate.

We have made the following revision:

“Using smRNA-FISH, we further mapped Olfr558 and Il12a transcripts to a separate subset of EC cells expressing Cpb2 (Fig. 4b,c), supporting the idea that there are subpopulations of EC cells in the proximal colon with gene transcripts associated with different physiological roles.”

(5) What are the white signals in the overlay in Fig5a, given that the Piezo1 probe (white) apparently did not give any staining by itself? Please consider a positive control for the Piezo1 probe.

The white signals in the overlay are Piezo1 staining that we do observe at what we consider background levels (also visible in the single-channel image).

(6) "Systematic administration of DT led to lethality in the Piezo2-DTR mice within 12 hours, but not in the Rosa26LSL-DTR or Piezo2-cre mice (data not shown), likely due to the essential function of Piezo2 in respiration" - presumably this should be corrected to "Systemic administration ...".

Thanks, this has been corrected to "Systemic administration ...".

(7) "Although gastric emptying (GE) was not affected in the Piezo2-DTR animals after DT treatment, small intestine transit (SIT) time, a measurement to assess the motility of small intestine, presented a small but statistically significant slowdown in the former group (Fig. 6g,h), suggesting that some Piezo2+ cells in the small intestine were depleted." - alternatively there could, of course, be a slowing of SIT in response to slower colonic transit independent of small intestinal epithelial Piezo2 or 5HT - to me this seems more likely given that even proximal colonic cells are spared in Fig6c and this should be discussed.

Thanks, that is a good point. We have made an amendment, which is shown in response to reviewer 1.

(8) In the context of the Villin-Cre experiments it should be discussed that other colonic EECs although express Piezo2, which might contribute to the observed phenotypes.

In our study, 97.7% of Piezo2+ cells in the distal colon had detectable Tph1 expression, suggesting that there is not a significant degree of overlap with other EEC types.

(9) MC4R is several times referred to as a nutrient-sensing moeity (e.g. in the discussion: "...and receptors associated with nutrient sensing (Casr and Mc4r), ...") - whilst the melanocortin system is important for nutrient homeostasis, MC4R is itself not a "nutrient sensor", a term usually reserved for the detection of macronutrients, such as amino acids, fatty acids, and monosaccharides; please reformulate.

We have amended this to “nutrient sensing and homeostasis”.